# ERRORRADAR: BENCHMARKING COMPLEX MATHEMATICAL REASONING OF MULTIMODAL LARGE LANGUAGE MODELS VIA ERROR DETECTION

## ABSTRACT

As the field of Multimodal Large Language Models (MLLMs) continues to evolve, their potential to handle mathematical reasoning tasks is promising, as they can handle multimodal questions via cross-modal understanding capabilities compared to text-only LLMs. Current mathematical benchmarks predominantly focus on evaluating MLLMs' problem-solving ability, yet there is a crucial gap in addressing more complex scenarios such as error detection, for enhancing reasoning capability in complicated settings. To fill this gap, we formally formulate the new task — **multimodal error detection**, and introduce **ERRORRADAR**, the first benchmark designed to assess MLLMs' capabilities in such a task. ERROR-RADAR evaluates two sub-tasks: *error step identification* and *error categorization*, providing a framework for evaluating MLLMs' complex mathematical reasoning ability. It consists of 2,500 high-quality multimodal K-12 mathematical problems, collected from real-world student interactions in an educational organization, with expert-based annotation and metadata such as problem type and error category. Through extensive experiments, we evaluated both open-source and closed-source representative MLLMs, benchmarking their performance against educational expert evaluators. Results indicate challenges still remain, as GPT-4o with best model performance is still around 10% behind human evaluation.

## 1 INTRODUCTION

On the path to Artificial General Intelligence, Large Language Models (LLMs) such as GPT-4 (OpenAI, 2023) have emerged as a central focus in both industry and academia (Minaee et al., 2024; Zhao et al., 2023; Zhu et al., 2023). As the real world is inherently multimodal, the evolution of Multimodal Large Language Models (MLLMs) such as the latest GPT-4o (OpenAI, 2024a) and Gemini 1.5 (Reid et al., 2024), has become a rapidly growing area of interest, demonstrating remarkable effectiveness in diverse applications (Xiao et al., 2024; He et al., 2024a; Yan et al., 2024; Hao et al., 2024). In particular, multimodal reasoning stands to significantly benefit education scenarios from the robust capabilities of MLLMs (Wang et al., 2024b; Li et al., 2024a), given its reliance on multimodal inputs to comprehensively grasp users' intentions and needs.

Within the multimodal sphere, mathematical scenarios pose a significant challenge, demanding sophisticated reasoning abilities from MLLMs (Lu et al., 2022; Ahn et al., 2024). These scenarios have attracted considerable research aimed at pushing the boundaries of MLLMs' reasoning capabilities (Hu et al., 2024; Jia et al., 2024; Lu et al., 2024c; Shi et al., 2024b; Zhuang et al., 2024). Besides, various representative benchmarks have been designed to measure MLLMs' performance in complex mathematical reasoning tasks, which involve multi-step reasoning and quantitative analysis within visual contexts (Lu et al., 2024b; Zhang et al., 2024; Qiao et al., 2024; Peng et al., 2024).

Scrutinizing the off-the-shelf mathematical reasoning benchmarks, there is a predominant focus on evaluating the problem-solving capabilities of MLLMs, prioritizing the accuracy with which MLLMs can solve mathematical problems (Wang et al., 2024a; Lu et al., 2024b; Zhang et al., 2024), as depicted in Figure 1 (a). However, in educational contexts, it is even more crucial to consider user-oriented needs, such as **error detection**. As indicated in Figure 1 (b), this involves not only pinpointing the first incorrect step in a student's step-by-step solution but also categorizing the types

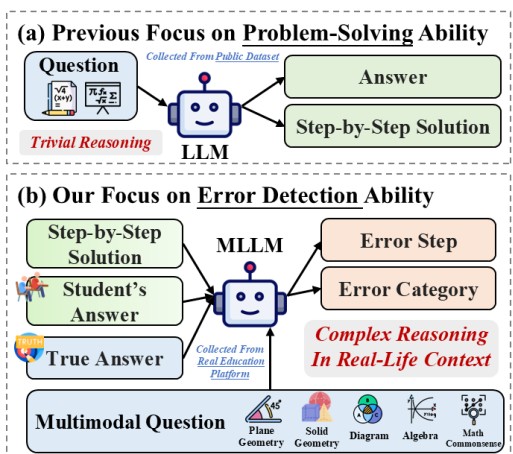

**Figure 1:** Comparison of research scope between previous work and our proposed ERRORRADAR benchmark on mathematical reasoning tasks.

| Benchmarks | Venue | Modality | Student Ans. | Error Det. |
|---|---|---|---|---|
| TheoremQA (Chen et al., 2023a) | EMNLP | $T$ | - | - |
| MathBench (Liu et al., 2024b) | ACL | $T$ | - | - |
| MR-GSM8K (Zeng et al., 2024) | arXiv | $T$ | - | - |
| SciEval (Sun et al., 2024) | AAAI | $T$ | - | - |
| EIC (Li et al., 2024b) | arXiv | $T$ | - | ✓ |
| CMMaTH (Li et al., 2024c) | arXiv | $T, I$ | - | - |
| MathScape (Zhou et al., 2024) | arXiv | $T, I$ | - | - |
| MATH-V (Wang et al., 2024a) | arXiv | $T, I$ | - | - |
| QRData (Liu et al., 2024c) | ACL | $T, I$ | - | - |
| IsoBench (Fu et al., 2024) | COLM | $T, I$ | - | - |
| SciBench (Wang et al., 2024c) | ICML | $T, I$ | - | - |
| MathVista (Lu et al., 2024b) | ICLR | $T, I$ | - | - |
| MathVerse (Zhang et al., 2024) | ECCV | $T, I$ | - | - |
| ERRORRADAR (Ours) | - | $T, I$ | ✓ | ✓ |

**Table 1:** Comparison between our proposed ERROR-RADAR benchmark vs. its relevant LLM-based mathematical reasoning benchmarks or datasets. Under the column of *Modality*, the letters $T$ and $I$ represent text and image, respectively. The column labeled as *Student Ans.* indicates whether the dataset contains real student data (*i.e.*, students' incorrect answers); the column labeled as *Error Det.* represents whether evaluation includes the complex reasoning task of error detection.

of errors made, which is a multifaceted process that requires a deep understanding of both mathematical concepts and cognitive processes (Davies et al., 2021; Rabillas et al., 2023).

Towards this end, addressing the aforementioned research gap, we aim to formulate the new task of evaluating MLLMs in the context of error detection scenarios, and therefore introduce the corresponding benchmark termed **ERRORRADAR**. We have designed two sub-tasks to comprehensively assess the performance: *error step identification* and *error categorization*. To construct a rich and reliable dataset, we initially sourced a collection of multimodal K-12 level math problems from an educational organization and subsequently refined the dataset through rigorous manual annotation to ensure quality. In particular, we also collect real students' answers for each multimodal question for a relatively robust experimental setting, compared to other relevant benchmarks (as shown in Table 1). Furthermore, we categorized the dataset to better align with diverse needs as follows: **Problem types**: *plane geometry*, *solid geometry*, *diagram*, *algebra*, and *mathematical common sense*; and **Error categories**: *visual perception errors*, *calculation errors*, *reasoning errors*, *knowledge errors*, and *misinterpretation of the problem*. In summary, the ERRORRADAR comprises 2,500 high-quality instances derived from real-life problem-solving data, providing a foundational dataset to enhance the complex reasoning capabilities of MLLMs for the research community and industry.

For ERRORRADAR, we carry out an extensive experimental analysis to determine the proficiency in complex mathematical reasoning of various MLLMs. The evaluation encompasses both the latest open-source MLLMs (*e.g.*, InternVL2 (Chen et al., 2023b), LLaVA-NEXT (Liu et al., 2024a), CogVLM2 (Wang et al., 2023a)), and closed-source MLLMs (*e.g.*, GPT4-o (OpenAI, 2024a), Gemini Pro 1.5 (Reid et al., 2024), Claude 3.5 (Anthropic, 2024b)). Our focus was on *their error detection capabilities, specifically the identification of the erroneous step and the classification of the error type.* To establish a comparative human performance standard, we involved expert human educators who possess a graduate-level degree or higher qualifications. The results demonstrate that ERRORRADAR, covering cutting-edge topics such as MLLMs' complex reasoning, poses a significant challenge, with human evaluation for two error detection tasks achieving less than 70%.

From in-depth evaluation of representative MLLMs, we obtain the following findings: ❶ Closed-source MLLMs, particularly GPT-4o, consistently outperform open-source MLLMs in both sub-tasks, and show more balanced accuracy across different error categories; ❷ Weaker MLLMs exhibit an over-reliance on simpler categories, while stronger models handle complex tasks better; ❸ Both MLLMs and humans perform better on error step identification compared to error categorization, as localizing specific errors is inherently simpler than categorizing errors.

Our contributions can be summarized as follows:

❶ We take the **first step to formulate the multimodal error detection task**, and introduce a multimodal benchmark termed ERRORRADAR for evaluation. This benchmark serves as a standard operator for assessing the complex mathematical reasoning capabilities of the latest MLLMs.

❷ We meticulously curate an extensive dataset comprising approximately 2,500 high-quality instances with rigorous annotation and rich metadata derived from real user interactions in an educational organization. To the best of our knowledge, this is the first attempt to use real-world student problem-solving data to evaluate MLLMs, providing a protocol for future research on MLLMs' complex mathematical reasoning.

❸ Our comprehensive experimental evaluation of more than 20 MLLMs, both proprietary and open-source, highlight the substantial room for improvement (*i.e.*, 7%-15% in accuracy) in the complex mathematical reasoning capabilities, underscoring the necessity for further research.

## 2 RELATED WORK

**Benchmarks for Mathematical Reasoning.** Recent advancements in mathematical reasoning benchmarks have led to the development of both pure text and multimodal assessments (Lu et al., 2022; Wang et al., 2024a; Zheng et al., 2024; Huo et al., 2024). While datasets like GSM8K (Cobbe et al., 2021), MATH (Hendrycks et al., 2021), SuperCLUE-Math (Xu et al., 2024), and MathBench (Liu et al., 2024b) focus on text-based problems, the field has expanded to include multimodal benchmarks that introduce visual elements, pushing the boundaries of AI's mathematical understanding. For instance, MathVista (Lu et al., 2024b) evaluates AI's performance on visual math QA tasks; MATH-V (Wang et al., 2024a) focuses on multimodal mathematical understanding with competition-derived questions; MathVerse (Zhang et al., 2024) assesses visual diagram comprehension using CoT strategies; CMMU (He et al., 2024b) tests multi-disciplinary, multimodal math understanding with a broad range of Chinese-language questions; MathScape (Zhou et al., 2024) further advances the field by presenting longer, more complex, and open-ended multimodal problems; and MMMU(Yue et al., 2024) covers college-level knowledge including interleaved mathematical questions. The aforementioned benchmarks assess the mathematical reasoning capabilities of MLLMs by evaluating their problem-solving levels, but they overlook tasks based on the student's perspective, such as error detection, and thus fail to comprehensively evaluate the more complex role of current MLLMs. Therefore, we propose the ERRORRADAR benchmark, which is entirely based on real student response data to evaluate the proficiency of MLLMs in error detection tasks.

**Multimodal Large Language Models.** Generative foundation models such as GPT-4 (OpenAI, 2023), Claude (Anthropic, 2024b), and Gemini (Pal & Sankarasubbu, 2024) have significantly advanced various task solutions without fine-tuning (Cui et al., 2024; Yan & Lee, 2024; Zou et al., 2025; Zhong et al., 2024). Similarly, current open-source MLLMs, built on top of powerful LLMs, have also demonstrated promising potential in multimodal tasks such as image captioning (Yang et al., 2024) and visual question answering (Fan et al., 2024). For instance, LLaVA-NEXT (Liu et al., 2024a) proposed projecting visual embeddings, extracted by a pretrained vision encoder, into the word space through a single MLP layer, where LLMs like LLaMA, Vicuna, and Mistral are fine-tuned to understand these post-projection tokens. In a similar fashion, Phi3 (Abdin et al., 2024), DeepSeek-VL (Lu et al., 2024a), MiniCPM-V (Yao et al., 2024), ChatGLM (GLM et al., 2024), CogVLM (Wang et al., 2023a), Intern-VL (Chen et al., 2023b), Qwen-VL (Bai et al., 2023) and Yi-VL (Young et al., 2024) also utilize a projector (or adapter, shared compression layer, *etc.*) to align the visual embeddings extracted from a vision encoder with text embeddings, which are then concatenated and fed into LLM. Therefore, we propose ERRORRADAR, a comprehensive benchmark on a fine-grained evaluation of MLLMs' ability to detect errors based on students' answers and reasoning steps, thereby advancing the development of complex multimodal system.

## 3 THE ERRORRADAR DATASET

### 3.1 TASK FORMULATION

**Basic Setting.** In this task, we assess the model's ability to detect errors in mathematical problem-solving processes across multiple samples. Let $N$ denote the total number of samples in the evaluation set. For each sample $i \in \{1, 2, \ldots, N\}$, the input set $\mathcal{I}_i$ is defined as:

$$\mathcal{I}_i = \{Q_{\text{text},i}, Q_{\text{image},i}, A_{\text{correct},i}, A_{\text{incorrect},i}, \{S_{k,i}\}_{k=1}^{n_i}\},$$

where:

- $Q_{\text{text},i}$: the textual statement of the $i$-th problem.
- $Q_{\text{image},i}$: the image representation of the $i$-th problem.

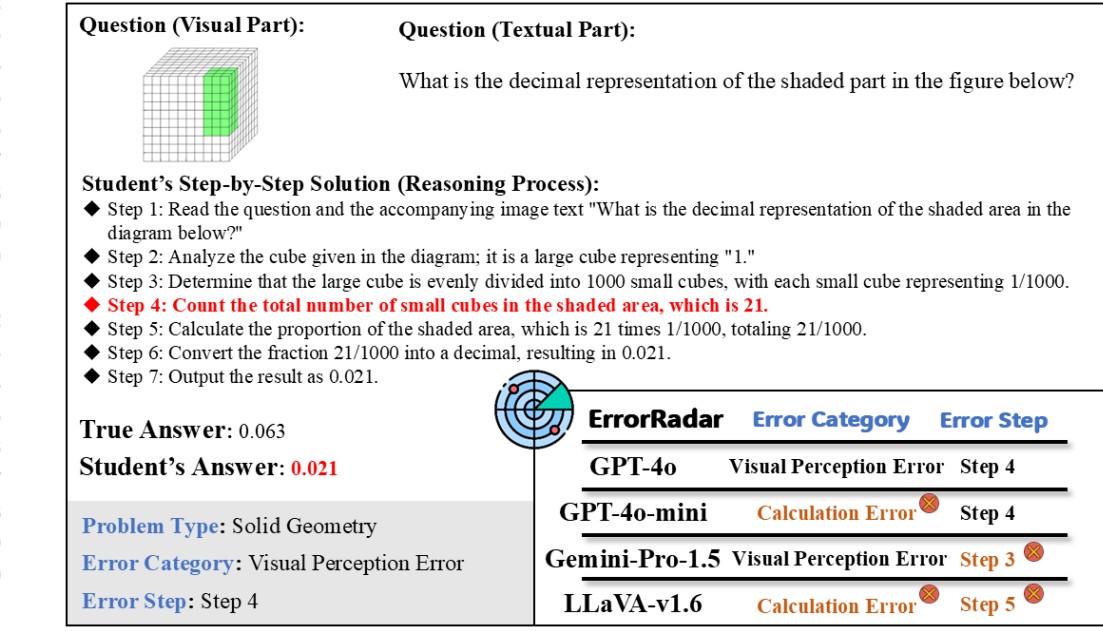

**Figure 2:** Example of our annotated multimodal mathematical reasoning dataset ERRORRADAR, and performance comparison on error categorization and error step localization tasks among representative MLLMs. It is evident that even simple math problems can be mishandled by the currently superior MLLMs in one or both tasks, highlighting the challenging nature of our proposed multimodal error detection setting.

- $A_{\text{correct},i}$: the correct solution for the $i$-th problem.
- $A_{\text{incorrect},i}$: the incorrect student solution for the $i$-th problem.
- $\{S_{k,i}\}_{k=1}^{n_i}$: the sequence of $n_i$ steps in the $i$-th problem-solving process, with each $S_{k,i}$ representing a distinct step.

**Subtask 1: Error Step Identification.** The task is to identify the index $x$ of the first incorrect step in the sequence $\{S_{k,i}\}_{k=1}^{n_i}$. The function $f_{\text{step},i}$ maps the input $\mathcal{I}_i$ to the index of the erroneous step:

$$f_{\text{step},i} : \mathcal{I}_i \rightarrow x_i, \quad \text{where } x_i = \arg\min_k \{S_{k,i} \text{ is incorrect}\}.$$

**Subtask 2: Error Categorization.** The task is to classify the type of error for the $i$-th problem into one of the following categories: $\{\text{VIS}, \text{CAL}, \text{REAS}, \text{KNOW}, \text{MIS}\}$. The error categorization function $f_{\text{error},i}$ maps the input $\mathcal{I}_i$ to the error category $C_{\text{error},i}$:

$$f_{\text{error},i} : \mathcal{I}_i \rightarrow C_{\text{error},i}.$$

More concrete examples can be seen in Figure 2 and Appendix A. The discrepancies within the five error categories are delineated as follows:

- ✫ **Visual Perception Errors (VIS)**: These errors arise when there is a failure to accurately interpret the information contained within images or diagrams presented in the question due to visual issues.

- ✫ **Calculation Error (CAL)**: These errors manifest during the calculation process, which may include arithmetic mistakes such as incorrect addition, subtraction, multiplication, or division, errors in unit conversion, or mistakes in the numerical signs between multiple steps.

- ✫ **Reasoning Error (REAS)**: These errors occur during the problem-solving process when improper reasoning is applied, leading to incorrect application of logical relationships or conclusions. In

- ✫ **Knowledge Error (KNOW)**: These errors result from incomplete or incorrect understanding of the knowledge base, leading to mistakes when applying relevant knowledge points.

- ✫ **Misinterpretation of the Question (MIS)**: These errors occur when there is a failure to correctly understand the requirements of the question or a misinterpretation of the question's intent, leading to responses that are irrelevant to the question's demands. For instance, if the question asks for a letter and a number is provided, or vice versa.

| Statistic | Number |
|---|---|
| Total multimodal questions | 2,500 |
| **Problem Type** | |
| - Plane Geometry | 1559 (62.4%) |
| - Solid Geometry | 191 (7.6%) |
| - Diagram | 233 (9.3%) |
| - Algebra | 288 (11.5%) |
| - Math Commonsense | 229 (9.2%) |
| **Error Category** | |
| - Visual Perception Error | 395 (15.8%) |
| - Calculation Error | 912 (36.5%) |
| - Reasoning Error | 951 (38.0%) |
| - Knowledge Error | 119 (4.8%) |
| - Misinterpretation of the Qns | 123 (4.9%) |
| Average Reasoning Step | 7.6 |
| Maximum Reasoning Step | 20 |
| Minimum Reasoning Step | 3 |
| Average Question Length | 168 |
| Maximum Question Length | 719 |
| Minimum Question Length | 13 |

**Table 2:** Key statistics of ERRORRADAR.

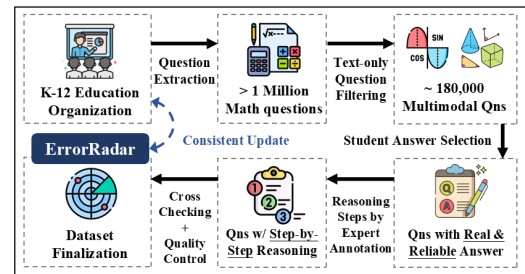

**Figure 3:** Roadmap of ERRORRADAR dataset collection, annotation, and consistent update.

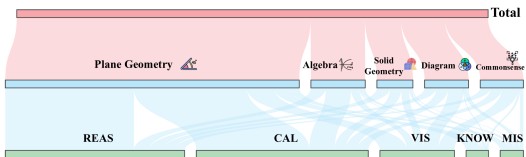

**Figure 4:** Dataset distribution of ERRORRADAR with respect to problem type and error category.

**Performance Metric.** The evaluation of both subtasks is conducted separately. The model's performance is evaluated using accuracy metrics for both subtasks:

- **Error Step Identification Accuracy.** Let $G_{\text{step},i}$ be the ground truth index of the first incorrect step for the $i$-th sample. The accuracy for this subtask is:

$$\text{Acc}_{\text{step}} = \frac{1}{N} \sum_{i=1}^{N} \mathbb{I}(x_i = G_{\text{step},i}),$$

where $\mathbb{I}(\cdot)$ is indicator function, returning 1 if prediction matches ground truth, and 0 otherwise.

- **Error Categorization Accuracy.** Let $G_{\text{error},i}$ be the ground truth error category for the $i$-th sample. The accuracy for this subtask is:

$$\text{Acc}_{\text{cate}} = \frac{1}{N} \sum_{i=1}^{N} \mathbb{I}(C_{\text{error},i} = G_{\text{error},i}).$$

## 3.2 DATA SOURCE & ANNOTATION

Following the roadmap shown in Figure 3, this section includes how we collect and annotate ER-RORRADAR dataset to ensure the overall data quality. Different from the conventional benchmarks that rely on public datasets or modified textbook collections (Lu et al., 2024b; Zhou et al., 2024), ERRORRADAR dataset is uniquely sourced from the question bank of a global educational organization. This repository encompasses a vast array of mathematical problems in K-12 levels, totaling over a million entries. Initially, we curated approximately 180,000 math problems that feature a single image-based question stem, aligning with our goal to target a multimodal assessment setup.

Subsequently, we refined our selection by evaluating the universality and articulation of the problem content. For each problem, we identified multiple incorrect answers. To ensure the dataset's relevance for error detection tasks, we selected the most frequently given incorrect answer as the student's response. Additionally, we scrutinized cases where the most common incorrect answer was due to system input errors despite the answer being correct. In such instances, we amended the dataset by incorporating the next most frequently incorrect answer.

Furthermore, since error detection tasks necessitate a step-by-step reasoning process, we enriched our dataset with new content through manual annotation. Specifically, we provided professional annotators with the original multimodal QA data, student's incorrect answers, and the pedagogical team's analysis of correct answer process. Based on this initial data, annotators delineated the erroneous steps leading to the incorrect answers (More details in Appendix B.1 and B.2).

Our team of annotators, consisting of around ten educational experts with domain expertise, conducted two rounds of cross-checking to ensure the reliability of the annotations. In cases of inconsistency, the contentious question and related data were presented to the annotation lead for final adjudication. The annotators' results were subject to review and quality control by the educational organization from which the data originated, ensuring security, reliability, and consistent updates.

## 3.3 DATASET DETAILS

As illustrated in Table 2, ERRORRADAR dataset comprises a substantial collection of 2,500 multimodal math questions designed for error detection tasks. It predominantly includes plane geometry problems, with solid geometry, diagram, algebra, and math commonsense questions making up the remainder, highlighting its focus on diverse mathematical problems. it also categorizes errors into visual perception, calculation, reasoning, knowledge, and question misinterpretation. Key statistics indicate a diverse dataset with an average reasoning step of 7.6, a variety of question lengths, and a wide range of reasoning steps (up to 38). Detailed distribution of ERRORRADAR, problem type definition, and error category formulation can be seen in Figure 4, Appendix B.3 and B.4.

## 4 EXPERIMENTS AND ANALYSIS

### 4.1 EVALUATION PROTOCOLS

In ERRORRADAR benchmark, we propose an evaluation strategy using template matching rules. The evaluation process consists of three stages: *response generation*, *answer extraction*, and *performance calculation*. Initially, the MLLMs generate responses given the inputs, which incorporates the multimodal mathematical question, wrong answer, and its step-by-step reasoning, using the template from Appendix C.2. Subsequently, the short answer text can be extracted from the detailed response. Finally, the model performance is based on the detailed score calculation as shown in Section 3.1. The final score will be calculated by averaging the scores from three rounds of assessment.

### 4.2 EXPERIMENTAL SETUP

In our experimental setup, we meticulously categorized and evaluated a diverse array of MLLMs into three distinct groups to assess their capabilities across error detection tasks. (i) The **Open-Source MLLMs** category encompassed models such as InternVL-2 (Chen et al., 2023b), Phi-3-vision (Abdin et al., 2024), Yi-VL (Young et al., 2024), DeepSeek-VL (Lu et al., 2024a), LLaVA-v1.6-Vicuna (Liu et al., 2024a), MiniCPM-LLaMA3-V2.5 (Yao et al., 2024), MiniCPM-V2.6 (Yao et al., 2024), Qwen-VL (Bai et al., 2023), GLM-4v (GLM et al., 2024), and LLaVA-NEXT (Liu et al., 2024a), each demonstrating their unique strengths and capabilities in handling different types of errors. (ii) The **Closed-Source MLLMs** featured proprietary models like Qwen-VL-Max (Bai et al., 2023), Claude-3-Haiku (Anthropic, 2024a), Claude-3.5-Sonnet (Anthropic, 2024b), Gemini-Pro-1.5 (Reid et al., 2024), GPT-4o-mini (OpenAI, 2024b), and GPT-4o (OpenAI, 2024a), providing a comparison point for the performance of models that are not publicly accessible. (iii) Lastly, the **Human Performance** category served as a benchmark for natural intelligence, allowing us to gauge how closely MLLMs can emulate human cognitive functions across tasks such as visual perception (More details in Appendix C.1). We provide the prompts for MLLMs and sources of MLLMs in Appendix C.2 and C.3, respectively.

### 4.3 EXPERIMENTAL RESULTS

#### 4.3.1 MAIN RESULTS

**Finding #1: Closed-source MLLMs generally outperform open-source MLLMs in both error detection tasks, with GPT-4o demonstrating the strongest performance.** Table 3 shows that closed-source MLLMs generally outperform open-source MLLMs in both STEP and CATE tasks, and they also exhibit relatively more balanced performance across the five error categories. This superiority can likely be attributed to the proprietary datasets and advanced training resources available to closed-source models, which allow for more robust fine-tuning (Shi et al., 2023; Yu et al., 2024; Wang et al., 2023b). Notably, GPT-4o stands out as the best model, achieving the highest scores not only in STEP and CATE tasks, but also in the VIS, REAS, and MIS categories, demonstrating its overall versatility and strength. Given the current performance gap between open-source

| Multimodal Large Language Models | Parameters | LLM | STEP | CATE | VIS | CAL | REAS | KNOW | MIS |
|---|---|---|---|---|---|---|---|---|---|
| *Open-Source MLLMs* | | | | | | | | | |
| InternVL2 (Chen et al., 2023b) | 2B | InternLM-2 | 9.8 | 25.1 | 32.2 | 38.8 | 12.2 | 0.0 | 24.4 |
| Phi-3-vision (Abdin et al., 2024) | 4B | Phi-3 | 37.5 | 40.7 | 9.6 | 99.6 | 6.6 | 3.4 | 4.1 |
| Yi-VL (Young et al., 2024) | 6B | Yi | 15.7 | 32.1 | 9.1 | 77.1 | 4.9 | 14.3 | 0.0 |
| DeepSeek-VL (Lu et al., 2024a) | 7B | DeepSeek | 16.2 | 35.7 | 4.6 | 90.9 | 0.4 | 28.6 | 6.5 |
| LLaVA-v1.6-Vicuna (Liu et al., 2024a) | 7B | Vicuna-v1.5 | 30.3 | 17.7 | 40.3 | 14.9 | 8.3 | 0.0 | 55.3 |
| InternVL-2 (Chen et al., 2023b) | 8B | InternLM-2.5 | 44.2 | 44.1 | 12.4 | 99.6 | 13.6 | 10.9 | 2.4 |
| MiniCPM-LLaMA3-V2.5 (Yao et al., 2024) | 8B | LLaMA3 | 37.4 | 38.0 | 4.1 | 100.0 | 2.1 | 2.5 | 0.0 |
| MiniCPM-V2.6 (Yao et al., 2024) | 8B | Qwen2 | 17.0 | 39.8 | 11.4 | 87.8 | 12.1 | 10.1 | 17.9 |
| Qwen-VL (Bai et al., 2023) | 9B | Qwen | 23.8 | 38.9 | 8.6 | 99.1 | 3.5 | 0.0 | 0.8 |
| GLM-4v (GLM et al., 2024) | 13B | GLM-4 | 44.6 | 44.1 | 2.5 | 92.9 | 25.8 | 0.0 | 0.0 |
| LLaVA-v1.6-Vicuna (Liu et al., 2024a) | 13B | Vicuna-v1.5 | 36.9 | 47.8 | 0.0 | 74.5 | 53.7 | 0.8 | 2.4 |
| CogVLM2-LLaMA3 (Wang et al., 2023a) | 19B | LLaMA3 | 15.0 | 20.1 | 43.3 | 33.8 | 0.7 | 13.4 | 0.0 |
| InternVL2 (Chen et al., 2023b) | 26B | InternLM-2 | 50.4 | 51.2 | 39.2 | 84.6 | 35.6 | 0.8 | 10.6 |
| LLaVA-NEXT (Liu et al., 2024a) | 72B | Qwen1.5 | 51.8 | 45.0 | 7.1 | 86.0 | 32.0 | 7.6 | 0.8 |
| InternVL2 (Chen et al., 2023b) | 76B | Hermes-2 Theta | 54.4 | 49.5 | 33.4 | 92.4 | 25.1 | 10.9 | 8.1 |
| *Closed-Source MLLMs* | | | | | | | | | |
| Qwen-VL-Max (Bai et al., 2023) | - | - | 48.7 | 52.9 | 15.2 | 78.9 | 50.5 | 14.3 | 36.6 |
| Claude-3-Haiku (Anthropic, 2024a) | - | - | 45.6 | 48.0 | 10.4 | 77.4 | 46.8 | 4.2 | 1.6 |
| Claude-3.5-Sonnet (Anthropic, 2024b) | - | - | 50.2 | 49.5 | 35.7 | 48.4 | 64.8 | 21.0 | 11.4 |
| Gemini-Pro-1.5 (Reid et al., 2024) | - | - | 55.0 | 52.7 | 43.5 | 55.7 | 63.1 | 18.5 | 13.0 |
| GPT-4o-mini (OpenAI, 2024b) | - | - | 52.0 | 44.5 | 9.1 | 46.8 | 62.7 | 31.9 | 13.0 |
| GPT-4o (OpenAI, 2024a) | - | - | 55.1 | 53.1 | 46.3 | 50.4 | 64.9 | 9.2 | 46.3 |
| *Human* | | | | | | | | | |
| Human performance | - | - | 69.8 | 60.7 | 66.8 | 75.9 | 47.6 | 35.3 | 53.7 |

**Table 3:** Comparison of open-source and closed-source MLLM performance (accuracy in percentage) across error detection tasks. We denote **STEP** and **CATE** for the performance of error step identification task (*i.e.*, $Acc_{step}$) and error categorization task (*i.e.*, $Acc_{cate}$), respectively. We also denote **VIS**, **CAL**, **REAS**, **KNOW**, and **MIS** for visual perception error, calculation error, reasoning error, knowledge error, and misinterpretation of the question. The highest and second highest scores (except for exceptional values) among MLLMs in each column are highlighted in red and blue, respectively. Exceptional values in CAL column are highlighted in grey, as more than 70% categories predicted by the MLLM are CAL (More analysis on Sec 4.3.1 Finding #2).

and closed-source MLLMs, open-source MLLMs can further enhance themselves by distilling the error detection capabilities of closed-source ones (Hsieh et al., 2023).

**Finding #2: Weak open-source MLLMs tend to predict CAL category, leading to unusually high performance.** Table 3 indicates that MLLMs with relatively low performance in the CATE task tend to exhibit unusually high performance in the CAL category. Specifically, open-source models like MiniCPM-LLaMA3-v2.5 even achieve a 100% accuracy in CAL, while Phi-3-vision and InternVL-2-8B reach 99.6%. Upon analyzing the category prediction proportions of CAL from Figure 5 (See details of all MLLMs in Appendix C.4), it becomes clear that open-source MLLMs with the top five CAL accuracy predict over 80% of instances as

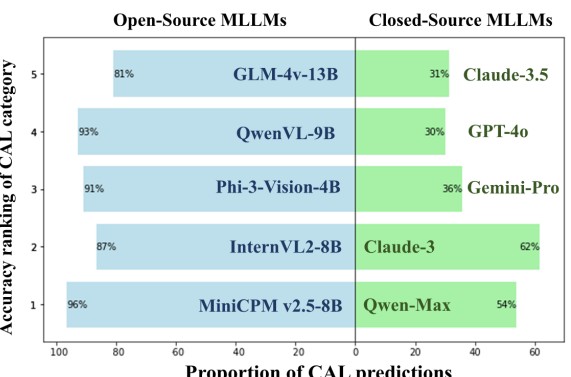

**Figure 5:** The proportion of CAL predictions of closed-source and open-source MLLMs with top-5 CAL accuracy.

CAL category, suggesting an over-reliance on this category. In contrast, closed-source MLLMs with top-five CAL accuracy do not exhibit this extreme trend of prediction bias. This phenomenon likely arises from weaker MLLMs attempting to overfit on the CAL category, a relatively simpler classification, to compensate for their inability to handle more complex scenarios (Tirumala et al., 2022; Xu et al., 2021). Models exhibiting this phenomenon can assign different weights to samples of different categories during training to reduce the model's preference for a particular category. This can be achieved by adjusting the weight parameters in the loss function (*e.g.*, Focal Loss & AdaFocal) (Li et al., 2022; Ghosh et al., 2022).

**Finding #3: MLLMs with strong overall performance tend to handle STEP easier than CATE.** From Table 3, the best open-source MLLMs, such as InternVL2-76B, and the best closed-source

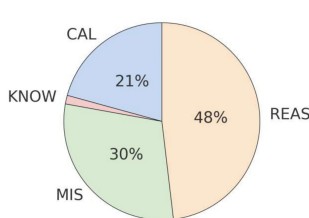

**Figure 6:** The error category distribution of misjudged VIS cases of GPT-4o.

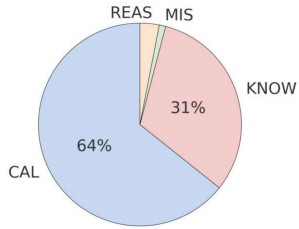

**Figure 7:** The error category distribution of misjudged VIS cases of CogVLM2-LLaMA3.

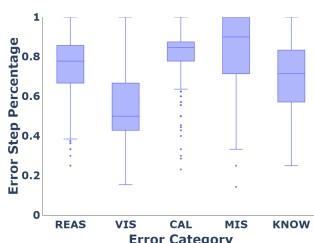

**Figure 8:** The error step distribution (in percentage) of error categories in ERRORRADAR.

MLLMs, like GPT-4o, exhibit a tendency where their STEP performance surpasses that of CATE. This trend holds even for human performance, where accuracy on STEP is higher (69.8%) compared to CATE (60.7%). The reason for this disparity is likely that identifying the error step is inherently easier, as it involves localizing a specific point of failure. On the other hand, categorizing the error requires more complex reasoning and contextual understanding to classify the nature of the error, which adds difficulty. This mirrors the settings in object detection, where localization (*i.e.,* predicting where an object is) is relatively simpler than classification (*i.e.,* predicting what an object is) (Zou et al., 2023; Jiao et al., 2021). To improve the performance of error categorization tasks, MLLMs need to better understand the relationship between the problem itself and the steps where errors occur. Thus, modeling this part of the relationship can be a focus in the design of training data (Ling et al., 2023; Shi et al., 2024a).

**Finding #4: CAL is the easiest category for MLLMs, while KNOW is the most difficult.** CAL is the category with the highest performance among most MLLMs (excluding those with exceptional values), which could be attributed to the structured and deterministic nature of calculations, where errors often result in clear, quantifiable deviations from expected outcomes, making them more straightforward to detect (Lewkowycz et al., 2022; Kojima et al., 2022). Conversely, KNOW stands out as the most challenging category, suggesting that MLLMs struggle significantly with tasks requiring deep factual understanding and contextual reasoning. The complexity of knowledge errors likely stems from the need for comprehensive domain expertise, which current MLLMs may not fully encapsulate yet. Even human performance reflects this trend, with knowledge error scoring notably lower than other categories, albeit with higher accuracy than MLLMs, highlighting the inherent difficulty of this task for both humans and AI (Kandpal et al., 2023; Feng et al., 2023). Thus, adding domain-specific knowledge to the dataset of MLLM is a direct solution (Ling et al., 2023).

**Finding #5: MLLMs still have a gap to close to reach human-level intelligence in error detection.** Human performance significantly outperforms the best MLLMs in both the STEP and CATE tasks, with accuracy scores of 69.8% and 60.7% respectively, compared to the highest MLLM scores of 55.1% and 53.1%. Notably, the detection of VIS by humans is markedly superior to the best MLLMs, with a difference of nearly 20%. This substantial lead may be attributed to the sophisticated pattern recognition inherent to human visual processing (Doerig et al., 2022), which MLLMs, despite their advancements, have yet to fully emulate. Besides, it is interesting to note that human performance in REAS detection is lower than all closed-source MLLMs but higher than almost all open-source MLLMs. This may suggest that closed-source MLLMs benefit from proprietary datasets and algorithms that better capture the nuances of logical reasoning (Wang et al., 2024d). To achieve human-level performance in error detection tasks, we can further introduce a reinforcement learning from human feedback (RLHF) approach, enabling the model to align with human thinking mechanisms in understanding error causes (Liu et al., 2023).

### 4.3.2 VISUAL PERCEPTION ANALYSIS

**Finding #1: Closed-source MLLMs are most likely to misjudge VIS as REAS in error categorization task.** Taking the best-performing GPT-4o model as an example, as shown in the Figure 6, 48% of VIS are misclassified as REAS, followed by 30% being misjudged as MIS. In multimodal mathematical scenarios, where the MLLM needs to handle information involving both visual and

linguistic elements simultaneously, particularly in problems related to plane and solid geometry, the complexity of the diagrams makes it difficult for the model to accurately extract certain features, leading to the frequent misclassification of VIS as REAS. For instance, if an erroneous response to a mathematical query originates from VIS (*e.g.*, misinterpreting a diagram), MLLM may mistakenly attribute this to a flaw in logical reasoning that occurs subsequent to initial visual misinterpretation.

**Finding #2: Open-source MLLMs are more likely to misclassify VIS as CAL.** Taking the open-source model CogVLM2-LLaMA3, which performs best in identifying VIS, as an example, CAL accounts for 64% of misclassified category, as illustrated in the Figure 7. When handling complex visual information, especially in geometry problems, the MLLM often struggles to accurately extract key features. Due to the open-source MLLM's weaker multimodal integration capabilities, it simplifies visual issues into numerical calculation problems. The lack of sufficient training and data for visual-related errors is also a key reason behind this phenomenon (Wichmann & Geirhos, 2023). More analysis on misclassification for each category can be seen in Appendix C.5.

### 4.3.3 RELATION BETWEEN ERROR CATEGORY AND ERROR STEP

**Finding #1: There is a close relationship between different error category and their distribution in the reasoning steps.** As shown in Figure 8, VIS tends to occur in the earlier to mid-stages, accounting for a median proportion of 0.5 of total steps. In contrast, MIS, REAS, CAL, and KNOW are more likely to arise in the later stages, with their median proportions ranging from 0.7 to 0.9. More analysis of this relationship across MLLMs can be seen in Appendix C.6.

**Finding #2: VIS occurs in the earlier stages of problem-solving reasoning**. This finding could be closely linked to the sequence in which students approach the task (Binz & Schulz, 2023; Kennedy & Romig, 2024). Since image content often serves as a key reference early on, any misinterpretation of this visual information directly impacts the subsequent problem-solving steps. Students typically first examine the image, and then integrate the information before proceeding to reasoning or calculation. As a result, visual perception errors arise earlier compared to other types of errors.

**Finding #3: Other error categories are primarily in later stages of problem-solving reasoning**. This may be linked to the increasing cognitive load students encounter during problem-solving. Cognitive Load Theory posits that information complexity ranges from low to high interactivity (Paas et al., 2010; Binz & Schulz, 2023). While low-interactivity information can be understood independently, high-interactivity information requires simultaneous processing of related elements, thus increasing cognitive load (Kennedy & Romig, 2024; Abbad-Andaloussi et al., 2023). In later stages, students must integrate complex information from multiple sources. For instance, calculating the distance between two points needs increasing interactivity heightens cognitive load, leading to errors like forgetting to take the square root or miscalculating differences. Consequently, as cognitive load rises, the frequency of errors in later steps also increases.

### 4.3.4 SCALING ANALYSIS

**Finding #1: The performance of MLLMs on STEP task increases with the scale of parameters.** We observe a phenomenon similar to the scaling law (Kaplan et al., 2020) in our experiments. As shown in Figure 9, when the size of the InternVL2 model increases from Tiny to Huge, the accuracy of STEP task rises from 9.8% to 54.4%, showing an improvement of 44.6%. Similarly, as the size of LLaVA-NEXT increases from Small to Large, its accuracy of STEP also improves from 30.3% to 51.8%, demonstrating that larger MLLMs exhibit greater reasoning ability in localizing erroneous steps.

**Finding #2: CATE task is relatively more difficult to improve through scaling.** While the accuracy of CATE shows a trend of improvement for both the InternVL2 and LLaVA-NEXT models as their size increases from Tiny (Small) to Middle, a slight decrease is also observed when

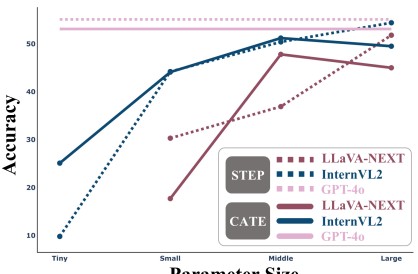

**Figure 9:** The accuracy of **STEP** and **CATE** of two representative MLLM series: LLaVA-NEXT and InternVL2. We denote *Tiny*, *Small*, *Middle*, *Large* as the 2B, 8B, 26B, 76B for InternVL2 and None, 7B, 13B, 72B for LLaVA-NEXT, respectively.

the model size reaches Large. We presume that this is because CATE is a more challenging task compared to STEP, and merely increasing the model size without fine-tuning is insufficient for

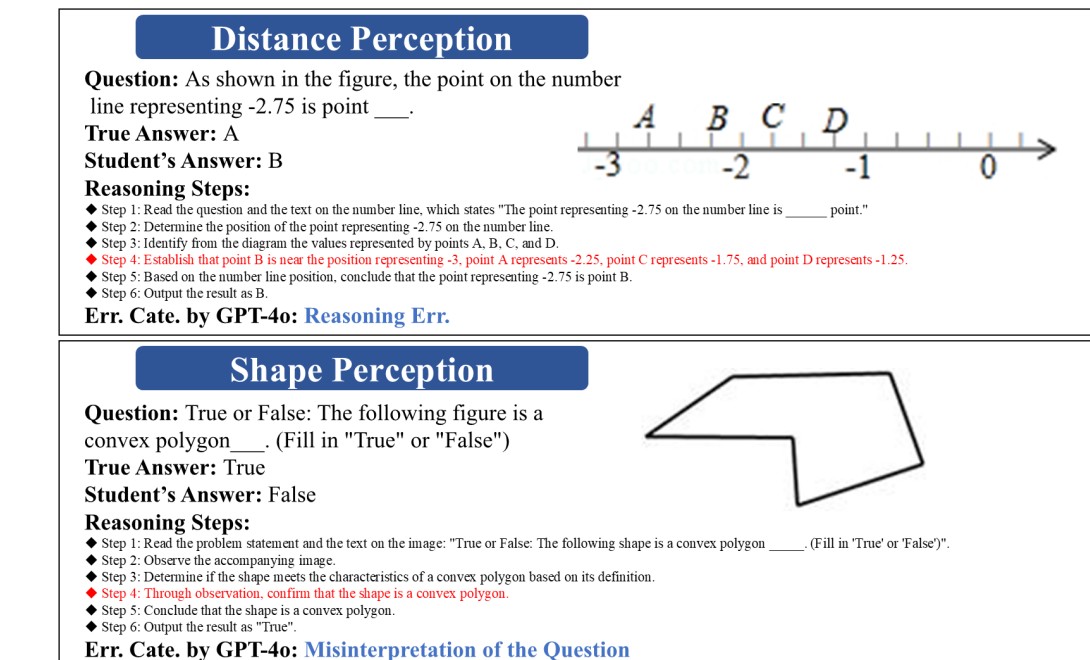

**Figure 10:** Bad cases where GPT-4o predicts visual perception errors incorrectly. Only distance and shape perception cases are shown here due to the page limit. More cases can be seen in Appendix C.7.

sustained improvement and may even introduce bias (Aghajanyan et al., 2023; Muennighoff et al., 2024). This phenomenon can also be seen in Table 3, consistent with Section 4.3.1 Finding #2.

### 4.3.5 VISUAL PERCEPTION CASE STUDY

Visual perception errors are critical in multimodal error detection tasks, as they impact the accurate comprehension of mathematical problems presented with both text and diagrams. As illustrated in Figure 10 and Appendix C.7, the five primary categories of visual errors observed in GPT-4o (the MLLM with best overall and VIS performance) include **distance perception**, **diagram perception**, **spatial perception**, **flip/fold perception**, and **shape perception**. These categories differ in their cognitive demands: distance perception focuses on point identification; diagram perception on quantitative estimation; spatial perception on geometric visualization; flip/fold perception on mental rotation; and shape perception on object classification (Lu et al., 2024b; Zhang et al., 2024). Detecting such errors is challenging because they often require both intricate visual processing and precise interpretation of mathematical relations, which can be difficult to encode in current MLLMs. To overcome these challenges, future MLLMs should incorporate more advanced visual reasoning capabilities, possibly through enhanced alignment between vision and language modalities, enabling better detection and correction of complex perception errors (Song et al., 2023). This could significantly improve the robustness of MLLMs in mathematical and other perception-heavy tasks.

## 5 CONCLUSION

In conclusion, this work introduces ERRORRADAR, the first multimodal benchmark designed specifically for evaluating MLLMs's reasoning in mathematical error detection scenarios. By focusing on both *error step identification* and *error categorization*, ERRORRADAR bridges a critical research gap in assessing MLLMs' capabilities in complex mathematical reasoning. The dataset's construction, based on real-world student interactions, ensures a robust evaluation framework that reflects genuine user needs. Our extensive experimental analysis, comparing leading open-source and proprietary MLLMs, reveals significant challenges in error detection, highlighting the need for continued advancements in this domain. As MLLMs continue to evolve, ERRORRADAR serves as an essential benchmark for driving improvements in the effectiveness of multimodal reasoning systems in real-world applications, on the path to Artificial General Intelligence.

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

## A    MORE MULTIMODAL QUESTION EXAMPLES

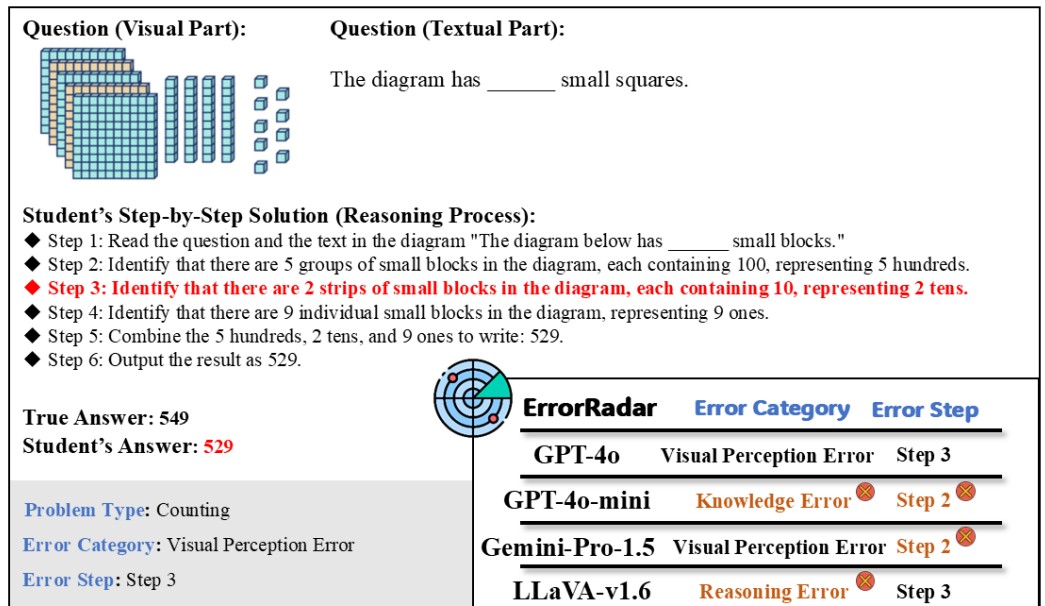

**Figure 11:** Multimodal mathematical example one (type: counting) from ERRORRADAR dataset.

**Question (Visual Part):**

**Question (Textual Part):**

The perimeter of the figure below is _______ cm.

**Student's Step-by-Step Solution (Reasoning Process):**
◆ Step 1: Read the text in the question and diagram "The perimeter of the figure below is _______ cm."
◆ Step 2: Determine the width of the figure as 20 cm.
◆ Step 3: Determine the length of the figure as 24 cm.
◆ **Step 4: Convert the perimeter of the figure into the perimeter of a rectangle with length of 24 cm and a width of 10 cm.**
◆ Step 5: Calculate the perimeter of the rectangle as 2 × (24 + 10).
◆ Step 6: Calculate the result as 2 × 34 = 68 (cm).
◆ Step 7: Output the result as 68.

**True Answer:** 88

**Student's Answer: 68**

**Problem Type:** Plane Geometry

**Error Category:** Reasoning Error

**Error Step:** Step 4

| ErrorRadar | Error Category | Error Step |
|---|---|---|
| **GPT-4o** | Reasoning Error | Step 4 |
| **GPT-4o-mini** | Visual Perception Error ⊗ | Step 4 |
| **Gemini-Pro-1.5** | Knowledge Error ⊗ | Step 4 |
| **LLaVA-v1.6** | Calculation Error ⊗ | Step 5 ⊗ |

**Figure 12:** Multimodal mathematical example two (type: plane geometry) from ERRORRADAR dataset.

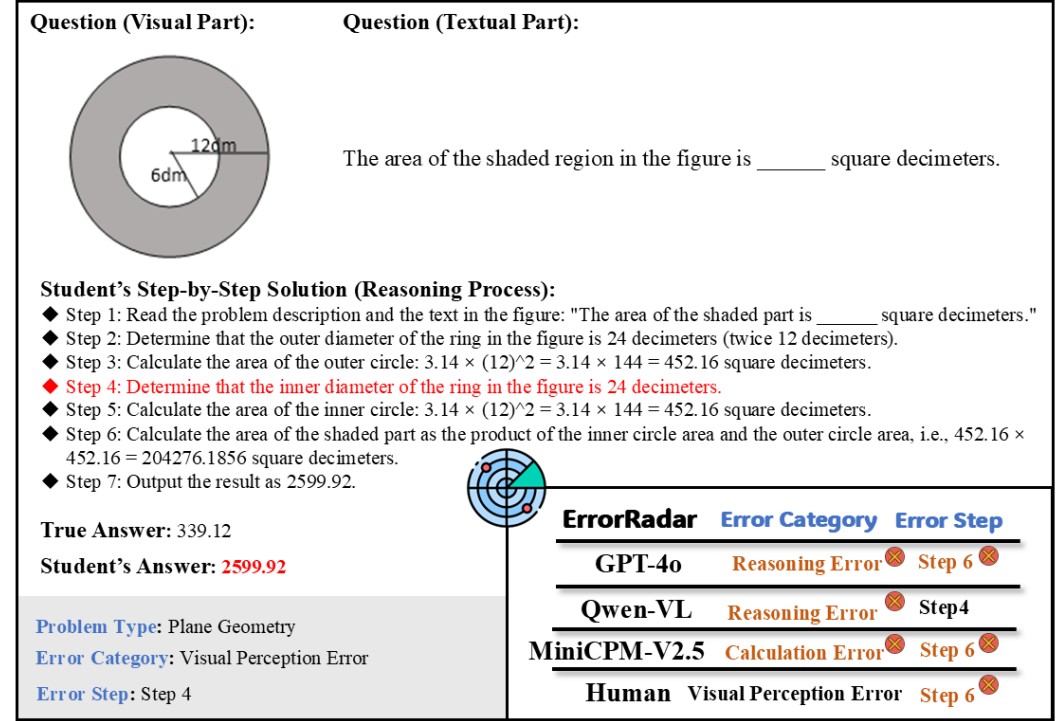

**Question (Visual Part):**

**Question (Textual Part):**

The area of the shaded region in the figure is _______ square decimeters.

**Student's Step-by-Step Solution (Reasoning Process):**
◆ Step 1: Read the problem description and the text in the figure: "The area of the shaded part is _______ square decimeters."
◆ Step 2: Determine that the outer diameter of the ring in the figure is 24 decimeters (twice 12 decimeters).
◆ Step 3: Calculate the area of the outer circle: 3.14 × (12)^2 = 3.14 × 144 = 452.16 square decimeters.
◆ Step 4: Determine that the inner diameter of the ring in the figure is 24 decimeters.
◆ Step 5: Calculate the area of the inner circle: 3.14 × (12)^2 = 3.14 × 144 = 452.16 square decimeters.
◆ Step 6: Calculate the area of the shaded part as the product of the inner circle area and the outer circle area, i.e., 452.16 × 452.16 = 204276.1856 square decimeters.
◆ Step 7: Output the result as 2599.92.

**True Answer:** 339.12

**Student's Answer: 2599.92**

**Problem Type:** Plane Geometry

**Error Category:** Visual Perception Error

**Error Step:** Step 4

| ErrorRadar | Error Category | Error Step |
|---|---|---|
| **GPT-4o** | Reasoning Error ⊗ | Step 6 ⊗ |
| **Qwen-VL** | Reasoning Error ⊗ | Step4 |
| **MiniCPM-V2.5** | Calculation Error ⊗ | Step 6 ⊗ |
| **Human** | Visual Perception Error | Step 6 ⊗ |

**Figure 13:** Multimodal mathematical example three (type: plane geometry) from ERRORRADAR dataset.

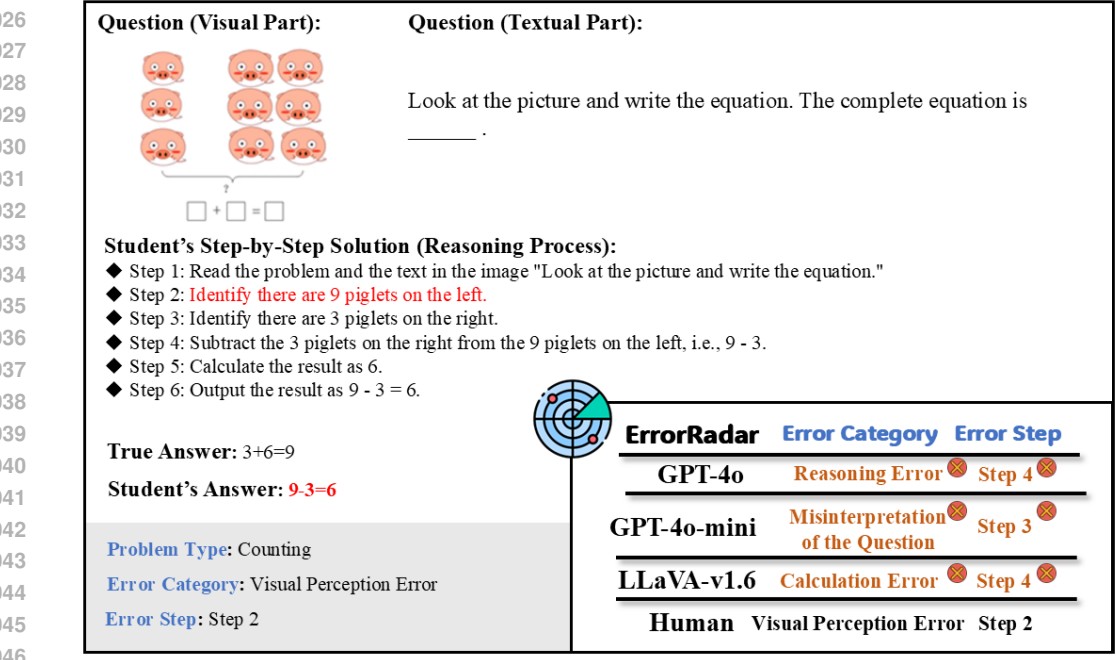

**Figure 14:** Multimodal mathematical example four (type: counting) from ERRORRADAR dataset.

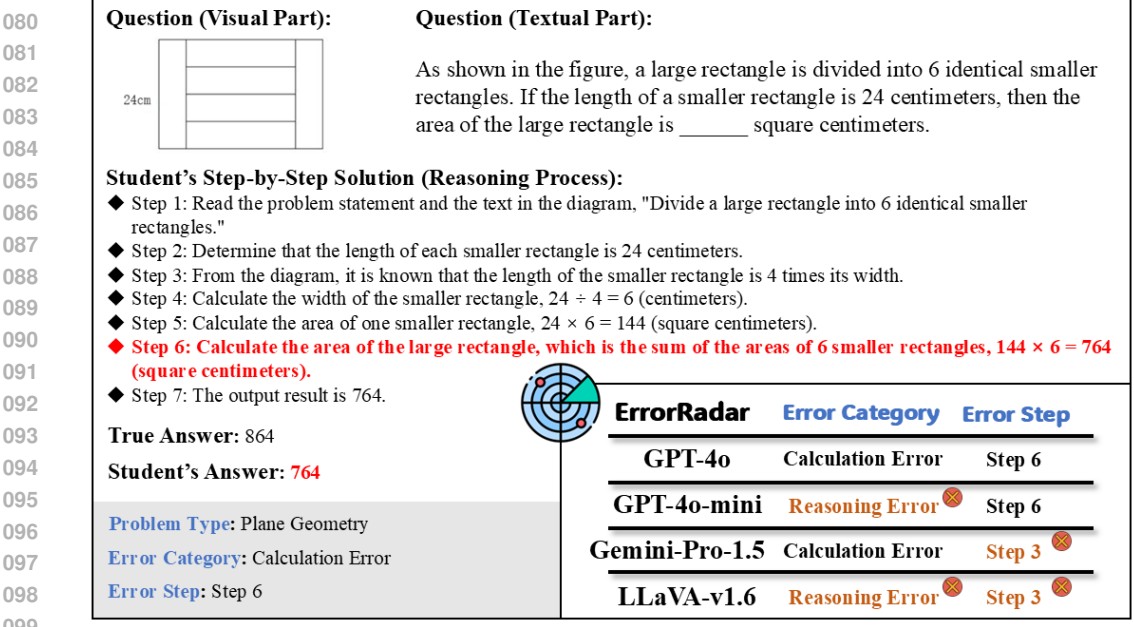

**Figure 15:** Multimodal mathematical example five (type: plane geometry) from ERRORRADAR dataset.

# B ADDITIONAL DATASET DETAILS

## B.1 ANNOTATION DETAILS

To ensure the quality and relevance of the ERRORRADAR dataset for error detection tasks, we employed a rigorous manual annotation process, involving professional educational experts as annotators. This section outlines the details of the annotation procedure, focusing on how the data was enriched with step-by-step reasoning processes, identification of erroneous steps, and error categorization.

**Annotator Selection and Training.** Given the complexity of the task, we recruited a group of ten annotators with specialized knowledge in educational theory and mathematics, particularly in K-12 pedagogy. These annotators were trained extensively to familiarize themselves with the structure and expectations of the task. The training covered the specifics of multimodal problem-solving in mathematics, typical student error patterns, and the need for precise identification of reasoning steps that led to incorrect answers. The annotators were also briefed on using the provided tools and the quality assurance process.

**Annotation Process.** Each mathematical problem in the dataset was annotated with a step-by-step reasoning process, capturing both correct and incorrect approaches to problem-solving. Annotators were provided with:

1. The original question stem (comprising both text and image components).
2. The student's most frequent incorrect answer.
3. The correct answer to the question.
4. The pedagogical analysis of the correct reasoning process, prepared by educational experts.

Based on these inputs, annotators were tasked with:

1. **Step-by-Step Reasoning Annotation**: For each problem, annotators mapped out the logical steps that students should ideally follow to arrive at the correct answer. This involved identifying key stages in the problem-solving process, such as formula application, arithmetic operations, or logical deductions.
2. **Error Step Identification**: For problems where students provided incorrect answers, annotators identified the exact steps where the reasoning went wrong. These error steps were explicitly marked and linked to the incorrect responses, ensuring that they could be traced back to specific problem-solving mistakes.
3. **Error Categorization**: Once the erroneous step was identified, annotators assigned an appropriate error category based on a predefined schema. These categories included common types of errors such as misinterpretation of the question (More details can be seen in Section 3.1). The categorization was designed to align with known student error patterns in mathematical learning.

**Quality Control and Cross-Validation.** To ensure annotation accuracy and consistency, each problem underwent two rounds of cross-checking:

1. **First Round of Cross-Validation**: After the initial annotation, another annotator independently reviewed the annotations. Any discrepancies between the first and second annotators were flagged for further analysis.
2. **Second Round of Cross-Validation**: In the second round, if the errors or discrepancies persisted, the problem was escalated to a senior educational expert who acted as the annotation lead. The annotation lead adjudicated these contentious cases, ensuring that the final decision was both pedagogically sound and aligned with the dataset's goals.

**Dataset Refinement.** Throughout the annotation process, we worked closely with the educational organization from which the dataset originated. This collaboration ensured that the annotations were not only reliable but also adhered to the standards of the organization's question bank. Additionally, ongoing feedback and updates from the organization helped refine the dataset, making it more accurate and relevant for multimodal error detection tasks.

**Annotation Duration and Effort.** The annotation process for the ERRORRADAR dataset spanned over a period of at least two months. During this time, the annotators, comprised of both professional educational experts and domain specialists, worked meticulously through several stages of preparation, annotation, and validation. Each annotator dedicated significant time to understanding the dataset, reviewing the provided pedagogical analyses, and applying their domain knowledge to identify and categorize errors. The first phase, involving step-by-step reasoning annotation, took approximately six weeks, while the subsequent cross-validation and quality control efforts accounted for the remaining two weeks. Given the complexity of the tasks and the necessity for high precision, the team's sustained efforts ensured that the final dataset was of the highest quality.

By incorporating these annotations, ERRORRADAR provides a robust foundation for studying student errors in mathematical reasoning and enables the development of advanced models for error detection and correction.

### B.2 DETAILS OF HANDLING INCONSISTENT ANNOTATIONS

To ensure the quality and reliability of our dataset for the multimodal mathematical error detection task, we established a systematic approach to resolve annotation inconsistencies. This process balances annotator independence with rigorous quality control, ensuring that the dataset is both accurate and representative.

#### B.2.1 ANNOTATION AGREEMENT PRINCIPLES

1. **Guided Consensus**: Annotations must align with clear, predefined guidelines covering the five error categories. Annotators are trained extensively to reduce subjective biases.

2. **Cross-Checking and Agreement Threshold**: Each instance is annotated by at least three annotators. Disagreements are flagged for further review.

3. **Systematic Review Process**: For inconsistent cases, a multi-step resolution process is applied:

   (a) **Initial Review**: Annotators discuss disagreements, referencing annotation guidelines and the specific problem context.

   (b) **Expert Arbitration**: For unresolved cases, a domain expert (e.g., an educational professional) reviews and finalizes the annotation.

   (c) **Consensus-Driven Decisions**: When possible, annotations are harmonized based on majority opinion or shared agreement after discussions.

#### B.2.2 CASE RESOLUTION FRAMEWORK

**Case Example 1: Visual Perception vs. Reasoning Error**

- **Example**: A problem presents a bar chart requiring students to determine the highest value. A student misidentifies the tallest bar and selects the wrong answer.
  - Annotator A labels this as a Visual Perception Error, arguing the mistake stems from misreading the chart.
  - Annotator B classifies it as a Reasoning Error, interpreting the mistake as a failure to compare values logically.

- **Resolution: Annotators revisit the problem:**
  - If evidence shows the student misunderstood the chart format (e.g., interpreting height as quantity but misjudging due to poor visualization), it is classified as a Visual Perception Error.
  - If the student correctly interprets the chart but misapplies logical comparisons (e.g., failing to compare values explicitly), it is categorized as a Reasoning Error.

  For persistent disagreement, an expert examines the student's work, including any notes or intermediate steps, to determine the correct annotation.

**Case Example 2: Knowledge vs. Misinterpretation of the Question**

- **Example**: A problem asks for the perimeter of a rectangle, but the student calculates the area instead.
    - Annotator A identifies this as a Knowledge Error, attributing the mistake to a lack of understanding of perimeter concepts.
    - Annotator B labels it as a Misinterpretation of the Question, asserting that the student misunderstood what was being asked.
- **Resolution**:
    - Did the student's work demonstrate understanding of the concept but apply it incorrectly (Misinterpretation of the Question)?
    - Did the mistake reveal a fundamental gap in knowledge about perimeter (Knowledge Error)?

    If disagreement persists, the annotators consult the expert, who may analyze additional context (e.g., previous responses or annotations).

### B.2.3 HANDLING IRRECONCILABLE DISAGREEMENTS

If discrepancies persist despite review and arbitration, the affected data points are excluded from the dataset. This strict policy prioritizes the overall quality and consistency of the dataset, ensuring that retained samples maintain high reliability.

### B.2.4 MONITORING AND FEEDBACK

Periodic feedback sessions are conducted to recalibrate annotators and refine guidelines based on observed patterns of disagreement. This iterative approach minimizes future inconsistencies and enhances annotator alignment over time.

## B.3 DEFINITION OF PROBLEM TYPE CATEGORY

The ERRORRADAR dataset distinguishes five primary types of multimodal mathematical problems, each characterized by unique features:

✶ **Plane Geometry Problems**: These involve two-dimensional shapes and figures, requiring knowledge of properties such as angles, lines, and polygons. Solving these problems often depends on understanding basic geometric principles and theorems about plane figures.

✶ **Solid Geometry Problems**: In contrast to plane geometry, solid geometry involves three-dimensional objects, such as cubes, cylinders, and spheres. These problems require spatial visualization and understanding of volume, surface area, and the relationships between different three-dimensional shapes.

✶ **Diagram-Based Problems**: These require analysis of provided visual information, such as graphs, charts, or diagrams, to solve mathematical queries. Interpreting visual data correctly is crucial, as these problems test the ability to extract and analyze quantitative information from visual aids.

✶ **Algebra Problems**: Algebra problems focus on abstract symbols and variables to represent numbers and relationships. These include tasks like solving equations, manipulating algebraic expressions, and understanding functions. The problem-solving process typically involves logical reasoning and manipulation of mathematical symbols.

✶ **Math Commonsense Questions**: These encompass a variety of problem types, including time judgment, direction judgment, counting, and pattern recognition. Unlike the other categories, math commonsense challenges rely on everyday mathematical reasoning and problem-solving strategies that do not necessarily require formal mathematical knowledge, testing intuitive understanding rather than procedural skills.

These problem types highlight the ERRORRADAR dataset's diverse nature, with each category presenting distinct challenges and requiring specific reasoning abilities.

### B.4 Development and Validation Process of Error Category

#### 1. Cross-Team Collaboration to Align Task Needs

The process began with close collaboration between the research team and the education team to ensure that the error categories aligned with the unique requirements of the multimodal math error detection task. The research team provided insights into the task's technical objectives, focusing on precision and comprehensive error coverage. Simultaneously, the education team contributed their understanding of real-world educational scenarios, emphasizing the practical relevance and applicability of the error taxonomy to students' and teachers' needs.

**Key Outcomes:**

- Initial consensus that the categories must address both multimodal challenges and real-life classroom scenarios.
- Recognition of the need to balance academic rigor with user-friendly categorization.

#### 2. Benchmark Survey and Focus Analysis

The research team conducted an extensive survey of representative benchmarks, focusing on error analysis frameworks in existing datasets. Examples included studies on problem-solving steps in educational AI and cognitive error modeling in multimodal tasks. The aim was to identify gaps in current frameworks and understand how existing taxonomies handle errors specific to visual, textual, and logical reasoning elements.

**Key Outcomes:**

- Identification of inadequacies in current benchmarks, particularly in addressing multimodal interactions like visual misinterpretations and reasoning errors tied to diagram-based tasks.
- Validation of the necessity for distinct categories to capture errors unique to multimodal math problems.

#### 3. Collection of Feedback from Students and Teachers

The education team collected qualitative and quantitative feedback from students and teachers to ensure that the proposed error categories were grounded in real-world educational needs. Focus groups, surveys, and interviews were used to gather perspectives on common error patterns encountered during classroom activities and assessments.

**Key Insights:**

- Teachers highlighted frequent calculation errors (**CAL**) and reasoning errors (**REAS**) as significant roadblocks to effective problem-solving.
- Students often reported confusion stemming from visual misinterpretations (**VIS**) and misunderstanding the question intent (**MIS**).
- Feedback emphasized the importance of separating reasoning-based errors from knowledge-based errors (**KNOW**) for better diagnostic support.

#### 4. Second Round of Discussion and Alignment

Following the feedback collection, the research and education teams reconvened to refine and align the error taxonomy. This phase involved iterative discussions to ensure that each category was distinct, comprehensive, and intuitive for annotators and end-users.

**Adjustments Made:**

- Clarified the scope of **Reasoning Errors (REAS)** to focus on improper logical application rather than factual knowledge gaps.
- Strengthened the definition of **Visual Perception Errors (VIS)** to address multimodal-specific challenges, such as interpreting diagrams or image-based data.
- Enhanced examples for each category to support annotation clarity.

## 5. Initial Finalization and Feedback from Educational Organization

The refined error categories were presented to a partner educational organization for feedback. This organization, which specializes in global education assessments, conducted an independent review and provided expert input.

**Key Outcomes:**

- Positive validation of the categories' relevance and comprehensiveness.
- Minor recommendations, such as specifying units and signs in the **Calculation Errors (CAL)** category, were integrated.

## 6. Final Validation and Alignment with Annotation Team

After incorporating feedback, the final set of error categories was finalized. The annotation team, comprising educational experts, received detailed guidelines and training to ensure consistent application of the taxonomy during the annotation process. Mock annotations were conducted to test the clarity and usability of the categories.

**Final Adjustments:**

- Annotators highlighted the need for clearer boundaries between **Reasoning Errors (REAS)** and **Knowledge Errors (KNOW)**, leading to additional examples and decision rules in the annotation guidelines.
- Alignment meetings ensured that all discrepancies and ambiguities were resolved before the dataset's official annotation began.

The aforementioned development process ensured that the five categories are comprehensive, robust, and applicable to both multimodal tasks and real-world educational scenarios.

# C  ADDITIONAL EXPERIMENTAL DETAILS

## C.1  HUMAN PERFORMANCE EVALUATION

In the Human Performance section, the evaluation involved three educational expert evaluators, each independently performing the two subtasks — error step identification and error categorization — on a set of multimodal math problems. To ensure the validity of their assessments, a rigorous cross-checking procedure was implemented. After the initial independent evaluations, the results from all three experts were compared for both the identification of error steps and the categorization of those errors. When discrepancies arose, particularly in cases where the experts disagreed on which step contained an error or how an error should be classified, a structured conflict resolution process was followed.

The cross-check process began with identifying areas of disagreement between the evaluators. These conflicts were discussed in a series of consensus meetings, where the evaluators would review the conflicting steps or categorizations in detail. Each expert provided their rationale, referencing the mathematical principles involved as well as the multimodal representations of the problems. Through open dialogue, the evaluators aimed to reach a consensus on the correct interpretation of the error.

In cases where consensus could not be easily achieved, a majority-vote system was employed. However, for particularly complex or ambiguous cases, an additional adjudicator — who did not participate in the initial evaluations but had equivalent expertise — was consulted to provide a final judgment. This adjudicator reviewed the contentious cases along with the evaluators' justifications, ensuring an unbiased final decision. The outcome of this process was the creation of a refined ground truth dataset that balanced expert knowledge with the goal of consistent and reliable error identification and categorization.

## C.2 PROMPT FOR MLLM EVALUATION

> **Task Definition:** You are an education expert proficient in K-12 mathematics. Your task is to identify the first step where the mistake occurred in the incorrect answer reasoning steps based on the following mathematical question (including the textual and visual parts), reference answer, and incorrect answer.
>
> **Output format:**
> Error Step: Step X
>
> **Below is the reference content you need to identify the error step:**
> Question Image: {image}
> Question text: {content}
> Correct Answer: {answer}
> Incorrect Answer: {user_answer}
> Incorrect Answer Reasoning Steps:{user_answer_steps}
>
> **Instruction:** Please provide the corresponding error step identification in the format "Error Step: Step X", without any additional content.

**Figure 16:** Prompt for error step identification task.

**Task Definition:** You are an education expert proficient in K-12 mathematics. Your task is to identify the category of error for the incorrect answer based on the following question (including the textual and visual parts), reference answer, and incorrect answer. The error should belong to one of the following categories: Visual Perception Error, Reasoning Error, Knowledge Error, Calculation Error, or Misinterpretation of the Question.

**Output format:**
Error Category: Clearly indicate which error category it belongs to.

**The definitions of the error categories are as follows:**
✶Visual Perception Error: Failure to accurately obtain information from the images or charts in the question due to visual issues, leading to errors.
✶Reasoning Error: Improper reasoning during the problem-solving process, failure to correctly apply logical relationships or draw conclusions, leading to errors
✶Knowledge Error: Errors occur when applying relevant knowledge points due to incomplete or incorrect understanding of knowledge.
✶Calculation Error: Errors occur in the calculation process, such as addition, subtraction, multiplication, division mistakes, or unit conversion errors, or errors in numerical symbols between multiple steps.
✶Misinterpretation of the Question: Failure to correctly understand the requirements of the question or misinterpreting the meaning of the question stem, leading to an irrelevant answer, such as answering with numbers when letters are required, and vice versa.

**Below is the reference content you need to identify the error step:**
Question Image: {image}
Question text: {content}
Correct Answer: {answer}
Incorrect Answer: {user_answer}
Incorrect Answer Reasoning Steps:{user_answer_steps}

**Instruction:** Please provide the corresponding error category in the format "Error Category: X", without any additional content.

**Figure 17:** Prompt for error categorization task.

## C.3 MODEL SOURCES

Table 4 details specific sources for the various MLLMs we evaluated. The hyperparameters for the experiments are set to their default values unless specified otherwise.

| MLLMs | Source | URL |
|---|---|---|
| InternVL2-2B | local checkpoint | https://huggingface.co/OpenGVLab/InternVL2-2B |
| InternVL2-8B | local checkpoint | https://huggingface.co/OpenGVLab/InternVL2-8B |
| InternVL2-26B | local checkpoint | https://huggingface.co/OpenGVLab/InternVL2-26B |
| InternVL2-76B | local checkpoint | https://huggingface.co/OpenGVLab/InternVL2-Llama3-76B |
| Phi-3-vision-4B | local checkpoint | https://huggingface.co/microsoft/Phi-3-vision-128k-instruct |
| Yi-VL-6B | local checkpoint | https://huggingface.co/01-ai/Yi-VL-6B |
| DeepSeek-VL-7B | local checkpoint | https://huggingface.co/deepseek-ai/deepseek-vl-7b-chat |
| LLaVA-v1.6-Vicuna-7B | local checkpoint | https://huggingface.co/llava-hf/llava-v1.6-vicuna-7b-hf |
| LLaVA-v1.6-Vicuna-13B | local checkpoint | https://huggingface.co/llava-hf/llava-v1.6-vicuna-13b-hf |
| LLaVA-NEXT-72B | local checkpoint | https://huggingface.co/llava-hf/llava-next-72b-hf |
| MiniCPM-V2.5-8B | local checkpoint | https://huggingface.co/openbmb/MiniCPM-Llama3-V-2_5 |
| MiniCPM-V2.6-8B | local checkpoint | https://huggingface.co/openbmb/MiniCPM-V-2_6 |
| Qwen-VL-9B | local checkpoint | https://huggingface.co/Qwen/Qwen-VL-Chat |
| GLM-4v-13B | local checkpoint | https://huggingface.co/THUDM/glm-4v-9b |
| CogVLM2-19B | local checkpoint | https://huggingface.co/THUDM/cogvlm2-llama3-chat-19B |
| Qwen-VL-Max | qwen-vl-max-0809 | https://modelscope.cn/studios/qwen/Qwen-VL-Max |
| Claude-3-Haiku | claude-3-haiku | https://www.anthropic.com/api |
| Claude-3.5-Sonnet | claude-3-5-sonnet | https://www.anthropic.com/api |
| Gemini-Pro-1.5 | gemini-1.5-pro-latest | https://deepmind.google/technologies/gemini/pro/ |
| GPT-4o-mini | gpt-4o-mini-2024-07-18 | https://platform.openai.com/docs/models/gpt-4o-mini |
| GPT-4o | gpt-4o-2024-08-06 | https://platform.openai.com/docs/models/gpt-4o |

**Table 4:** Sources of our evaluated MLLMs.

## C.4 CAL AND NON-CAL DISTRIBUTION OF MLLMS

In this section, we indicate the distribution of CAL and non-CAL category predictions of 21 MLLMs we evaluate, as shown in Figure 18. It can be seen that there is a bias towards CAL category among most open-source MLLMs, while closed-source ones except for Claude-3-Haiku and Qwen-VL-Max do not have such a bias for error categorization task.

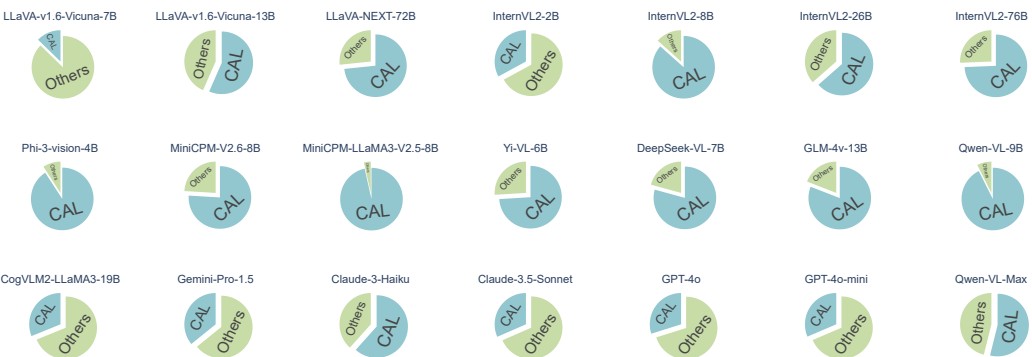

**Figure 18:** Distribution of CAL and non-CAL category predictions of all MLLMs we evaluate.

## C.5 ANALYSIS OF CONFUSION MATRIX FOR CATE TASK

Figures 19 and 20 present the confusion matrices for InternVL2-76B and GPT-4o, two MLLMs evaluated on five error categories. The matrices show the count of predictions for each category, with diagonal entries representing correct predictions and off-diagonal entries indicating misclassifications. These visualizations provide insights into each model's strengths and weaknesses.

InternVL2-76B shows strong performance in detecting CAL, with 843 correct predictions, indicating its robust numerical reasoning capability. However, the model struggles to distinguish between REAS and CAL, misclassifying 626 REAS instances as CAL. This confusion suggests an over-reliance on numerical features and an inability to separate logical reasoning tasks from computational ones. Additionally, there is significant misclassification of VIS into CAL, with 244 cases, highlighting a potential weakness in integrating visual and textual modalities. These trends may stem from InternVL2-76B's limited domain-specific reasoning ability.

GPT-4o, on the other hand, demonstrates relatively good performance in VIS, with 183 correct predictions, significantly outperforming InternVL2-76B. Its capability in REAS is also notable, with 617 correct predictions, suggesting a more balanced reasoning ability. However, GPT-4o struggles more with CAL, achieving only 460 correct predictions, and shows significant confusion between CAL and REAS, with 299 CAL instances misclassified as REAS. Furthermore, the model has difficulty with MIS, misclassifying 45 MIS cases as REAS, pointing to challenges in identifying nuanced interpretational issues. These trends suggest that GPT-4o's emphasis on multimodal alignment and contextual understanding contributes to its strengths in VIS and REAS but comes at the expense of CAL performance.

Comparing the two models reveals distinct strengths and weaknesses. GPT-4o significantly outperforms InternVL2-76B in VIS, likely due to superior multimodal visual-text alignment capabilities. Both models exhibit confusion between REAS and CAL, but GPT-4o shows a more balanced classification ability in REAS. MIS remains a challenging category for both models, though GPT-4o struggles slightly more in distinguishing it from REAS. These differences may arise from variations in model architecture and training objectives. This analysis underscores the complementary strengths of these models: InternVL2-76B excels in numerical reasoning, while GPT-4o performs better in visual perception and logical reasoning. Future research could explore ways to integrate their strengths for a more robust multimodal error detection system.

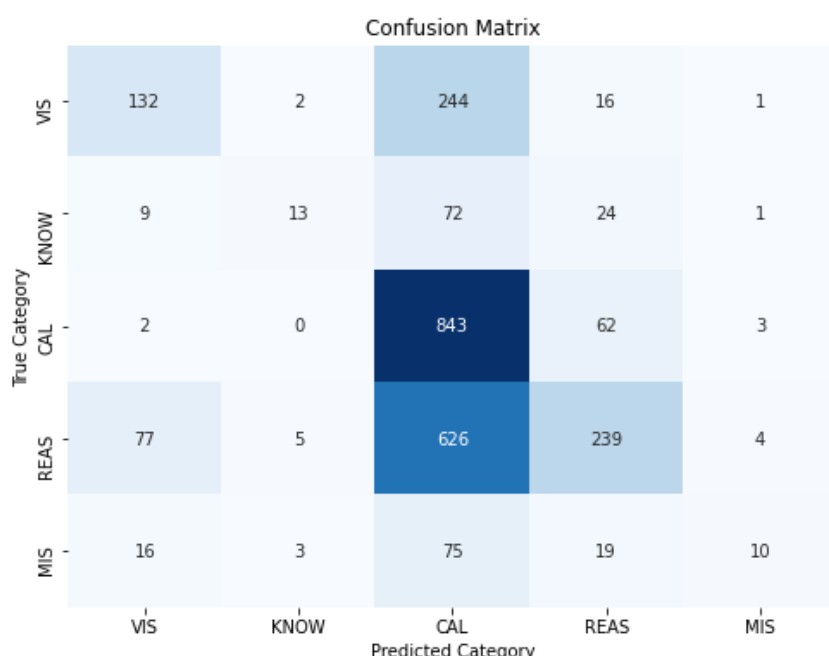

**Figure 19:** The confusion matrix of five error categories predicted by InternVL2-76B, the open-source MLLM with the best overall performance on error detection.

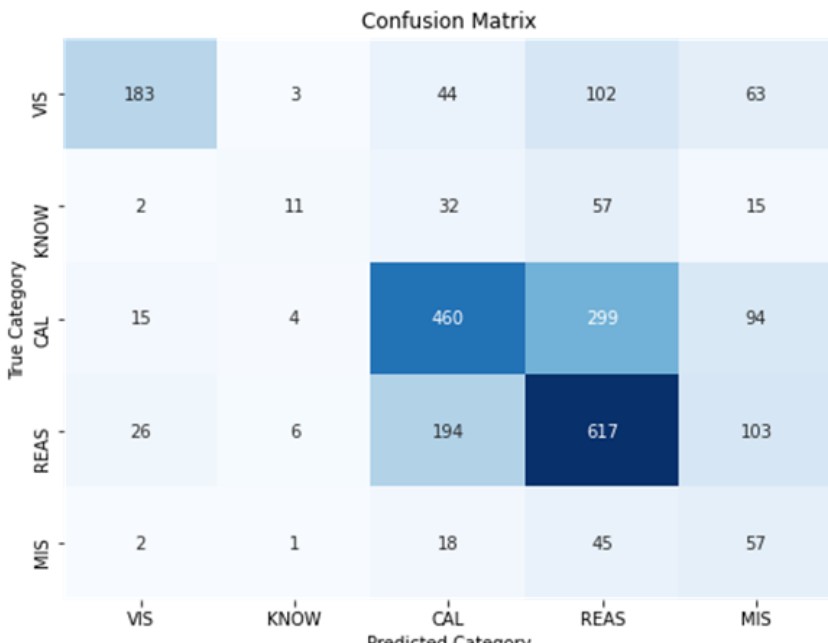

**Figure 20:** The confusion matrix of five error categories predicted by GPT-4o, the closed-source MLLM with the best overall performance on error detection.

## C.6  COGNITIVE LOAD ANALYSIS ACROSS MLLMS

In analyzing the error step distribution for the multimodal error detection task using InternVL2-76B (see Figure 21) and GPT-4o (see Figure 22), we observe a consistency in the pattern of error category distribution across both MLLM's predictions and those in ERRORRADAR (see Figure 8). In particular, VIS tends to occur in the earlier stages of problem-solving for both MLLMs, which aligns with the sequence in which students typically approach tasks. Since visual content often serves as a key reference at the outset, any misinterpretation of this information can significantly impact subsequent steps. Students generally examine the image first and then integrate the information before proceeding to reasoning or calculation, leading to visual perception errors arising earlier compared to other types of errors.

Other error categories, such as REAS, CAL, MIS, and KNOW, are more likely to emerge in the later stages of problem-solving. This pattern is linked to the increasing cognitive load students encounter as they progress. According to Cognitive Load Theory, information complexity ranges from low to high interactivity. Low-interactivity information can be understood independently, whereas high-interactivity information requires the simultaneous processing of related elements, thereby increasing cognitive load. In the later stages, students must integrate complex information from multiple sources, which can lead to errors like forgetting to take the square root or miscalculating differences when calculating distances, for example. Consequently, the frequency of errors in later steps increases with the rising cognitive load.

Despite the overall pattern being consistent, there may be subtle differences between InternVL2-76B and GPT-4o in terms of error step distribution, especially for MIS category. These differences could be attributed to the models' distinct architectures and training data, which might influence their approaches to error detection. As an open-source MLLM, InternVL2-76B might not have been optimized for specific types of questions or educational contexts, which could lead to a higher variability in MIS.

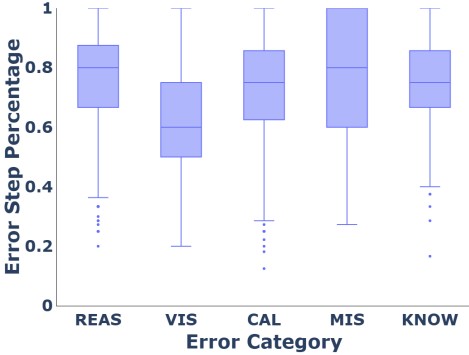 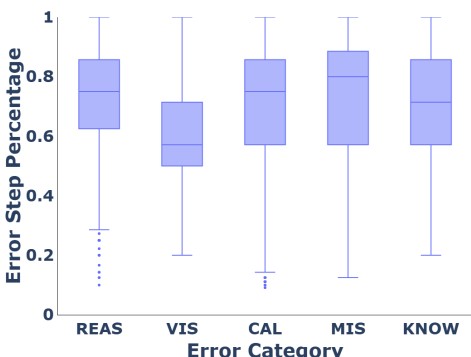

**Figure 21:** The error step distribution (in percentage) of error categories predicted by InternVL2-76B, the open-source MLLM with the best overall performance on error detection.

**Figure 22:** The error step distribution (in percentage) of error categories predicted by GPT-4o, the closed-source MLLM with the best overall performance on error detection.

## C.7 VISUAL BAD CASES PREDICTED BY GPT-4O

Figures 23, 24 and 25 show bad cases where GPT-4o predicts visual perception errors incorrectly, in terms of diagram, spatial and flip & fold perception.

---

### Diagram Perception

**Question:** To understand the growth situation of a certain type of flower in a botanical garden, a sample of 200 flowers was randomly selected from a plot with approximately 3000 such flowers, and their heights were measured. After organizing the statistical results, the data is listed as follows: (each group of data may include the minimum value but does not include the maximum value) Estimate the number of such flowers in the plot with a height less than 55 cm and not less than 45 cm to be about ______ flowers.

**True Answer:** 960
**Student's Answer:** 1440

| Height(cm) | 40~45 | 45~50 | 50~55 | 55~60 | 60~65 | 65~70 |
|---|---|---|---|---|---|---|
| Frequency | 33 | 42 | 22 | 24 | 43 | 36 |

**Reasoning Steps:**
◆ Step 1: Read the problem statement and chart text "Estimate the number of such flowers in the garden with a height less than 55 cm and not less than 45 cm is about ______ plants."
◆ Step 2: Determine the total sample size as 200 plants.
◆ Step 3: Determine the total number of flowers as 3000 plants.
◆ Step 4: Read from the chart that the number of flowers with a height of 45-50 cm is 42 plants.
◆ Step 5: Read from the chart that the number of flowers with a height of 50-55 cm is 32 plants.
◆ Step 6: Calculate the number of flowers with a height between 45-55 cm as 42 + 32 = 72 plants.
◆ Step 7: Calculate the proportion of flowers with a height between 45-55 cm in the sample as 72 / 200 = 0.36.
◆ Step 8: Multiply the total number of flowers, 3000 plants, by the proportion in the sample, 0.36, to calculate approximately 3000 × 0.36 = 1440 plants.
◆ Step 9: Output the result as 1440.
**Err. Cate. by GPT-4o: Calculation Err.**

**Figure 23:** Diagram bad case where GPT-4o predicts visual perception errors incorrectly.

---

### Spatial Perception

**Question:** Using small cubes with an edge length of 1 cm to form the geometric shape shown in the figure, place this shape on a table and paint the exposed surfaces with color (excluding the bottom surface), then the total area of the painted surfaces is ______ cm².
**True Answer:** 27
**Student's Answer:** 41
**Reasoning Steps:**
◆ Step 1: Read the problem statement and the text in the diagram, "The geometric shape shown is made up of small cubes with an edge length of 1 cm. Place this geometric shape on the table and paint the exposed surfaces (excluding the bottom surface), then the total area of the painted surfaces is ___1___."
◆ Step 2: Observe the structure of the geometric shape and confirm it is composed of small cubes.
◆ Step 3: From the front view, count the number of small squares, totaling 6.
◆ Step 4: From the left view, count the number of small squares, totaling 6.
◆ Step 5: From the top view, count the number of small squares, totaling 8.
◆ Step 6: From the right view, count the number of small squares, totaling 6.
◆ Step 7: From the back view, count the number of small squares, totaling 6.
◆ Step 8: Add the number of small squares from each face, totaling 6 + 6 + 8 + 6 + 6 = 32.
◆ Step 9: Count the number of small squares on the bottom face, totaling 9.
◆ Step 10: Add the number of small squares on the bottom face to the previous total, 32 + 9 = 41.
◆ Step 11: Calculate the total area of the painted surfaces, which is the area of 41 small squares.
◆ Step 12: Output the result as 41.
**Err. Cate. by GPT-4o: Misinterpretation of the Question**

**Figure 24:** Spatial bad case where GPT-4o predicts visual perception errors incorrectly.

---

### Flip/Fold Perception

**Question:** A number is shown in the mirror as illustrated, then the actual number is ______.
**True Answer:** 2502
**Student's Answer:** 2505

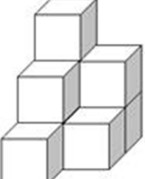

**Reasoning Steps:**
◆ Step 1: Read the problem statement and the text in the image: "The number in the mirror is shown in the figure."
◆ Step 2: Observe that the number in the mirror is "5025."
◆ Step 3: Based on the properties of mirror symmetry, the number seen in the mirror is actually flipped horizontally and vertically.
◆ Step 4: Flip the number "2502" horizontally to get the actual number "2505."
◆ Step 5: Output the result as 2505.
**Err. Cate. by GPT-4o: Reasoning Err.**

**Figure 25:** Flip & fold bad case where GPT-4o predicts visual perception errors incorrectly.

