# OpenReview forum: "ErrorRadar: Benchmarking Complex Mathematical Reasoning of Multimodal Large Language Models Via Error Detection"
_ICLR.cc/2025/Conference — Submitted to ICLR 2025_

### Official Review · Reviewer_XUMv · 2024-11-03

**Soundness:** 4
**Presentation:** 3
**Contribution:** 3
**Rating:** 6
**Confidence:** 3

**Summary:**

The paper,  introduces ERRORRADAR, a new benchmark designed to evaluate Multimodal Large Language Models (MLLMs) on their ability to detect errors in complex mathematical reasoning. Unlike traditional benchmarks that focus solely on problem-solving accuracy, ERRORRADAR addresses the challenge of identifying and categorizing errors in student solutions. It includes two main tasks: error step identification, which locates the first incorrect step in a student's solution, and error categorization, which classifies the error type.

The dataset comprises 2,500 annotated K-12 math problems, incorporating real student answers from a large educational organization and covering a range of error categories such as visual perception, calculation, reasoning, knowledge, and question misinterpretation. Extensive experiments comparing various open-source and closed-source MLLMs, including GPT-4o and Gemini 1.5, demonstrate that while MLLMs are improving, they still perform below human-level accuracy in error detection. ERRORRADAR provides a valuable resource for the research community, offering a structured approach to assess and enhance MLLMs' reasoning abilities in multimodal educational settings.

**Strengths:**

Originality:A primary strength of this paper is its originality, specifically in redefining evaluation for mathematical reasoning tasks. ERRORRADAR goes beyond traditional problem-solving benchmarks by focusing on error detection, introducing two distinct tasks: error step identification and error categorization. These tasks address a critical aspect of educational assessment that current benchmarks overlook, particularly when using multimodal data. By categorizing errors into five types—visual perception, calculation, reasoning, knowledge, and misinterpretation—the benchmark provides a nuanced approach that aligns with real-world educational needs. The use of real student responses adds further originality, as few existing benchmarks leverage authentic student data to assess model capabilities in complex reasoning.

Quality: The methodological quality of this paper is high, marked by rigorous data curation and detailed experimental design. The ERRORRADAR dataset consists of 2,500 well-annotated, multimodal math problems with stepwise incorrect student responses. The dataset is collected from a large educational organization, ensuring diversity and relevance to actual student problem-solving scenarios. The authors evaluate a comprehensive set of open-source and closed-source MLLMs and benchmark them against human evaluators, providing a robust comparison. This broad evaluation captures multiple facets of model performance, highlighting strengths and limitations across both error identification and categorization tasks. The inclusion of accuracy metrics for different models across error types offers valuable insights for improving MLLMs’ reasoning abilities.

Clarity: The paper is generally well-structured and clear in its presentation, with a logical progression from problem formulation to methodology and results. Figures, such as performance comparisons and error distribution visuals, effectively support the narrative, enhancing reader comprehension. The detailed explanation of the error types and tasks aids in understanding the benchmark’s scope and objectives. However, certain sections, particularly those describing the technical setup for error categorization, could benefit from further clarification, which would increase accessibility for a wider audience. Overall, the clarity of the main contributions is strong, though minor improvements could enhance the paper’s readability.

Significance: The significance of this work is substantial, as it opens new directions for evaluating MLLMs in educational contexts, where error detection is as crucial as problem-solving accuracy. By focusing on multimodal mathematical error detection, ERRORRADAR highlights a critical area where current models fall short of human performance, particularly in understanding errors in visual and logical reasoning. The findings emphasize the need for further research to bridge this gap, making the benchmark a valuable tool for both academia and industry. Given the growing use of AI in educational technology, ERRORRADAR’s contributions are timely and likely to stimulate further work in developing models that better support human learning and assessment.

**Weaknesses:**

Issue: The paper evaluates a broad range of MLLMs, yet the chosen models are not fully representative of the diversity in architectural design or training methodologies. The focus is largely on mainstream MLLMs, which may limit insights into the broader applicability of ERRORRADAR to varied model architectures.
Suggestion: Incorporating models that represent different architectural approaches, such as hybrid symbolic-neural systems or models with explicit reasoning layers, could offer a more comprehensive view of how ERRORRADAR evaluates different reasoning frameworks. This would help determine whether specific architectures consistently perform better on certain error types, which could have implications for future MLLM design.

Issue: While ERRORRADAR’s dataset is derived from real student interactions, the paper does not address how the benchmark might perform in actual deployment scenarios, such as in online tutoring systems or educational applications where students interact with AI in real-time.
Suggestion: The authors could simulate a real-world deployment setting, assessing factors such as response time, user feedback integration, or adaptability in evolving learning environments. This would add practical value, helping readers understand how ERRORRADAR might function outside of controlled benchmarking scenarios and providing insights into potential deployment challenges.


Issue: Different MLLMs may exhibit varying biases toward certain error categories due to their underlying training data and design, but the paper does not delve into these biases in detail. This oversight could affect the fairness and reliability of the models in educational contexts, where bias could disadvantage certain student demographics.
Suggestion: An analysis of category-specific biases in the models would add depth to the findings. For instance, are certain models disproportionately prone to labeling calculation errors over reasoning errors? Investigating these tendencies could inform guidelines for model selection and training adjustments to ensure balanced performance across error types.

Issue: The paper touches on cognitive load associated with different types of errors (e.g., reasoning errors appearing in later problem-solving steps), but it does not provide an in-depth analysis of how cognitive load might impact error detection performance across models.
Suggestion: Including a more detailed cognitive load analysis, possibly with additional metrics or models designed to handle high cognitive load tasks, would provide richer insights. For example, testing models’ performance on questions with high interactivity and cognitive demand could highlight specific areas where models may benefit from cognitive load-aware training.


Issue: While the dataset annotation process is described, details regarding inter-annotator agreement, consistency checks, and the process for resolving discrepancies are sparse. This information is crucial for understanding the reliability of the dataset, especially given the complexity of categorizing mathematical errors.
Suggestion: Adding inter-annotator agreement metrics, outlining quality control measures, and discussing how ambiguous cases were handled would strengthen the dataset’s credibility. It would also help readers gauge the dataset’s reliability for evaluating complex reasoning tasks.


Issue: The evaluation focuses on overall model performance, but lacks targeted baselines or ablation studies for each error type, especially for the multimodal aspects of error detection, which involve both textual and visual information. This limits the insights into specific error categories and multimodal model performance.
Suggestion: Introducing error-specific baselines, such as comparing models on tasks that isolate visual or reasoning errors, would clarify which aspects of ERRORRADAR are most challenging. Ablation studies that selectively disable visual or language components could also reveal the contribution of each modality to overall performance, guiding future multimodal benchmark development.

Issue: The paper does not address the computational demands of using ERRORRADAR to evaluate large models, which could impact its adoption in academic and industry settings with limited resources.
Suggestion: Discussing the benchmark’s computational requirements, particularly for larger MLLMs, and proposing strategies to improve efficiency (e.g., optimized sampling or reduced evaluation steps for preliminary testing) would make ERRORRADAR more accessible. Additionally, insights into potential trade-offs between accuracy and computational efficiency could benefit readers interested in practical deployment.

**Questions:**

Question: Did you consider evaluating ERRORRADAR with models beyond mainstream MLLMs, such as hybrid symbolic-neural models or models with dedicated reasoning layers? How well do you believe ERRORRADAR would generalize across different model architectures?
Suggestion: Including diverse model types could reveal whether certain architectures perform better on specific error categories, providing insights into optimal architectural approaches for error detection in complex reasoning tasks.


Question: How do you envision ERRORRADAR performing in real-world educational settings, such as in online tutoring or AI-driven assessment tools? Have you considered testing or simulating the benchmark under real-time conditions?
Suggestion: Running a real-world simulation or discussing any potential deployment challenges would provide readers with valuable context on how ERRORRADAR might function in practical applications, enhancing its relevance.

Question: Did you observe any biases in error detection, such as models favoring certain error categories? For instance, do any models show a tendency to overpredict certain types of errors?
Suggestion: Providing an analysis of category-specific biases could help address fairness concerns in educational applications. Understanding these biases could inform the development of more balanced models and benchmark settings.


Question: Can you provide further details on the annotation process, such as inter-annotator agreement scores or quality control measures? How were discrepancies handled, especially for complex or ambiguous cases?
Suggestion: Including inter-annotator agreement metrics and a description of the quality control process would improve confidence in the dataset’s reliability, especially given the nuanced nature of error categorization.

Question: Did you conduct any ablation studies or baseline evaluations focused on specific error types, such as isolating visual perception errors? How does each modality contribute to the overall performance of ERRORRADAR?
Suggestion: Performing ablation studies or creating error-specific baselines could clarify which modalities (textual vs. visual) or model components contribute most to performance, potentially guiding improvements in multimodal model design.

---

### Official Review · Reviewer_ekLk · 2024-11-03

**Soundness:** 2
**Presentation:** 2
**Contribution:** 3
**Rating:** 6
**Confidence:** 3

**Summary:**

This paper proposes a new benchmark for evaluating multimodal LLMs (MLLMs) on mathematical reasoning tasks, and in particular introducing the component of error detection, wherein models are evaluated on detecting where and what types of reasoning errors occur in solutions. These results are compared across several models (both closed and open), and with a baseline from human educators.

**Strengths:**

This work addresses a concrete gap, which is clearly established in Table 1: establishing a benchmark that is multimodal (image + text) and has error detection support. I also think several aspects of the analysis in this paper are interesting and valuable, e.g. comparison of several diverse models, attempts to have a comprehensive annotation process, and recruitment of examples reflecting various different topics (shown in stats in Table 2).

**Weaknesses:**

Overall, I think more work is needed to establish the robustness of the benchmark.

Given the relatively low performance of the human expert baselines, it’s worth analyzing what the human errors are due to. This would help to:
1. Ensure the low human performance is not due to label noise
2. Better calibrate our expectations of MLLMs

For example, I would not have assumed current generation models would beat or come close to strong human baselines on any of these tasks (given that human educators set the baselines across educational contexts, and that LLMs have well documented flaws in mathematical reasoning).

If human expert performance is really this low on such tasks, then there is a chance that it could be due to:
1. A major, unobserved flaw in our current mathematical education system
2. Low agreement between annotators, which might challenge the establishment of this benchmark as having strong ground truth annotations (if educators can plausibly disagree to this extent)

One way of ensuring #2 does not come into play would be to provide a detailed analysis of inter-annotator agreement. However, the consensus-based process described in Appendix C might not support this (unless the initial disagreements were recorded, and some metric of agreement could be computed from these). In any case, I strongly encourage the authors to provide some reasoning as to why the human performance is where it is, and some evidence that this is not due to factors like label noise, natural and plausible disagreement, etc. I acknowledge that model performance can be evaluated against the human noise/agreement ceiling to mitigate this, but such a low ceiling (e.g. 35.3% on KNOW) makes it a bit difficult to estimate the reliability of the model performance findings.

When thinking about the value of the paper’s findings to the community:
- It’s not clear to me that Finding #1 in the main results (i.e. 3.3.1) makes a significant contribution. It’s generally known that leading closed-source models (e.g. GPT-4o) outperform open models, and this is not surprising given the commercial incentives of model providers and the large cost to train them. Is there some nuance in the results that couldn’t be obtained without this contribution? If so, it would be helpful to highlight this here.
- The extremely high performance of relatively weak MLLMs on CAL doesn’t seem adequately explained by the hypothesis in the paper that such models overfit to this category to avoid learning more complex ones. If the category is indeed simple, then stronger models should perform well on it too, unless (intuitively) doing better on CAL tends to causally lower performance on other categories. So, I think a more compelling explanation of this phenomenon is needed to trust that such a result would generalize. For example, an alternative hypothesis is that these models have seen these or similar questions during their pretraining, trivially memorizing the questions or near neighbors, and the models that do not perform well on this category have not. I notice two papers are cited as to the paper’s hypothesis for the result; if there is some more compelling explanation relating to CAL overfitting, a clear explanation and allusion to those papers’ results would be very helpful.
- Likewise, the explanation for Finding #3 doesn’t seem complete. The analogy to object detection isn’t clear; a more direct analogy would be if localization (predicting where an object is) is simpler than classification (predicting what an object is). However, is this the case empirically? Categorization seems simpler than “fully understanding [objects’] attributes or behavior” since solving a classification task doesn’t seem to require this deeper understanding (e.g. we would not claim that an ImageNet-trained classifier understands the attributes or behavior of zebras, for example, but distinguishing them from horses seems possible via pattern recognition.
- The findings in section 3.2.2 generalize across closed-source vs. open MLLMs, but is there an a priori reason to believe that the openness of a model causally affects its capabilities or confusion on different tasks? Or is this more of a commentary on scale/performance in general? I.e. could a line be drawn between low-performing or high-performing models? This seems like it would agree with the scaling law analysis presented in 3.3.4. Otherwise, again it’s not clear that this interpretation of results is robust to e.g. a low-capability closed-source model (such as local models used in things like Apple Intelligence) or a high-capability open model (such as large LLaMa models).
- Additionally, the analysis in 3.2.2 might benefit from plotting a confusion matrix, rather than highlighting only the most common mis-associations in results (this could be in an appendix). Figures 6 and 7 show the erroneously selected categories, but not what they were supposed to be, which makes it challenging to observe what kinds of nuanced reasoning errors might be driving this behavior (and where models could be improved, more specifically).


Some notes on writing and presentation:
- The abstract obscures the core contributions, and also contains several subjective claims (e.g. “comprehensive”, “rigorous”, “rich” without definition of what is meant by these). It would be helpful to simplify it to focus on the factual contributions of this work.
- Similarly, the applicability of multimodal LLMs to mathematical reasoning tasks is cited as “especially” promising, but no reason is given for this (e.g. over and above text-only LLMs, which are largely used for mathematical tasks currently). I recommend making this clearer, since it would help readers quickly orient to the potential contribution of the given work.
- Line 096: What exactly does “featuring advanced research scopes such as complex reasoning” mean?
- Line 099: It is a bit odd for the paper to refer to its own contribution as insightful. This is a determination traditionally left to external reviewers, the community, etc. Such cases occur in other parts of the paper as well (e.g. in the abstract as noted previously, or like "well-annotated” in the Figure 2 caption). I encourage the authors to carefully use neutral language when discussing their results, and focusing on the facts established in the paper.
- Re: notation in 2.1, $T$ is used to denote problem, but this is a bit confusing (given that text is a subset of a problem, and text also begins with T). Wouldn’t it be clearer to use $P$ for problem, or $Q$ for question, or similar? Considering that $A$ is answer, $S$ is step $\mathcal{M}$ is model, etc.
- Similarly the use of $Step$ for an incorrect step, such that $Step \in S$ is a bit confusing, considering that this also seems to be denoted by $S_{x,i}$. Also, shouldn’t the output be simply $x$, i.e. the step index rather than the step content itself? It might be helpful to do a pass on notation to help simplify and make it more intuitive, so the reader can quickly follow.

**Questions:**

What are the reasons that human performance is so far from 100% (despite the source being human annotations), given that the annotators and participants seem to be educators?

What is the foundation for the 5 error types $C$? I.e. are they based on the authors’ intuition, heuristics from prior work, a survey of mathematics educators, etc.?

Can finding #1 be simply assumed by deduction from the generally better performance of closed-source models across tasks?

Is the dataset private, in the sense that it’s unlikely to have been seen by current generation MLLMs?

---

### Official Review · Reviewer_ywyz · 2024-11-04

**Soundness:** 2
**Presentation:** 3
**Contribution:** 3
**Rating:** 6
**Confidence:** 4

**Summary:**

The paper introduces ERRORRADAR, a benchmark for evaluating the complex mathematical reasoning of multimodal large language models (MLLMs) through error detection tasks. The key contributions include:

1. New Task Formulation: Establishes the multimodal error detection task, focusing on error step identification and error categorization.
2. Dataset Creation: Compiles a dataset of 2,500 K-12 multimodal math problems based on real student interactions, with detailed annotations and error categories.
3. Comprehensive Evaluation: Assesses various open-source and closed-source MLLMs, comparing their performance with human evaluators.
4. Insights and Challenges: Highlights the challenges MLLMs face in matching human-level performance, particularly in complex reasoning tasks.

Overall, ERRORRADAR provides a framework for improving the reasoning capabilities of MLLMs in educational settings.

**Strengths:**

1. The paper introduces a new benchmark for error detection in mathematical reasoning, which is a unique focus compared to existing benchmarks. The dataset is well-constructed from real-world student data, with rigorous annotation and a comprehensive evaluation of multiple models.
2. The authors did a comprehensive evaluation involving 21 MLLMs and provide some detailed analysis.
3. This work has the potential to significantly impact the development of educational AI systems by improving error detection capabilities.

**Weaknesses:**

Overall, this work proposes a good benchmark but has some issues, primarily related to the task introduction, the soundness of the experimental results, and the depth of the analysis.

1. The authors claim the benchmark addresses "more complex" mathematical reasoning, but the dataset contains only K-12 level questions. It would be beneficial for the authors to elaborate on how the error detection tasks for K-12 questions require complex reasoning, better with specific examples.

2. Although the authors provide the formulated task representation in section 2.1, it is not clear or correct enough. Specifically:
    1. From equations on line 188 and 199, The input and output for the two subtasks are not clearly defined. It lacks clarity on whether the question text, image, correct answer, and incorrect answer are part of the input for each of the subtask.
    2. The combined description of inputs and outputs for both subtasks (lines 149, 161, and 166) is confusing, suggesting the task requires training a model that complete the two subtasks simultaneously, which is not the case.
    3. The use of complex variables in Section 2.1 might be unnecessary and make the easy thing more complex, and could be simplified with natural language for better understanding.
Overall, I would suggest the author re-organizing the content in section 2.1 and make it clear and simple.

3. The result presentation and metrics for error categorization may have some issues.
    1. The metric used for reason-specific results resembles recall rather than accuracy. For error categorization, using precision, recall, and F1-score would better reflect performance on biased data and biased prediction. Could the authors provide results on these metrics and explain how they might change the interpretation of the results?
    2. The red value for CAL in Table 3 is not the highest, contrary to what is indicated.

4. The conclusion on line 355 cannot be supported. The accuracies of STEP are not consistently better than those of CATE (e.g., InternVL2 actually achieves much better scores on CATE instead), and they are not hugely different even for strong MLLMs in many cases. Additionally, the two subtasks have different numbers of classification types, which may influence the accuracy values.

5. More experimental settings can be considered, such as evaluating the impact of including or excluding correct/incorrect answers in STEP, or testing a pipeline approach where the output of STEP feeds into CATE.

6. While the analysis is detailed, it lacks deeper insights or actionable findings. For example, does the output of the models explain why models miscategorize errors? Are there some fundamental ability shortage of the models that we need to further improve? What can we expect and improve for the future applications of MLLMs in education? Etc. The authors may consider providing deeper analyses and discussion on these or other issues.

**Questions:**

1. Why does this benchmark have to focus on image-included samples? Does there exist some text-only benchmarks for text-only error detection?

2. How the authors identify and define the steps in the students' solutions?

3. The work only focuses on the first incorrect steps. Cannot we also consider all the incorrect steps? Would it be possible that the models actually identified other incorrect steps after the first one but were regarded as false predictions?

---

### Official Review · Reviewer_wmhi · 2024-11-07

**Soundness:** 2
**Presentation:** 2
**Contribution:** 2
**Rating:** 5
**Confidence:** 4

**Summary:**

This paper presents ERRORRADAR, an innovative benchmark aimed at evaluating the capabilities of Multimodal Large Language Models (MLLMs) in mathematical error detection, a new direction within mathematical reasoning. The authors identify a significant gap in current benchmarks, which tend to focus primarily on problem-solving, and address this by proposing a formalized task that emphasizes complex reasoning and error identification. ERRORRADAR comprises 2,500 high-quality K-12 math problems derived from real-world student interactions, with extensive metadata and rigorous annotation, making it a unique and robust contribution to the field.

The authors' experimental evaluation, involving both open-source and proprietary MLLMs, effectively highlights the challenges in this task, with models like GPT-4 still trailing human evaluators by approximately 10%. This emphasizes the limitations of current MLLMs in error detection and suggests a clear path for future research.

The strengths of this work lie in its originality, its real-world data foundation, and the thorough experimental analysis across a range of MLLMs. However, the benchmark’s long-term impact will depend on whether it can spur advancements in complex reasoning within MLLMs. Overall, this paper makes a meaningful contribution to the field and provides a promising resource for evaluating and improving MLLMs' mathematical reasoning capabilities.

**Strengths:**

Originality: The paper’s introduction of multimodal error detection in MLLMs is novel, as it addresses an important but understudied aspect of AI reasoning.

Quality:
ERRORRADAR’s dataset is substantial, with 2,500 rigorously annotated K-12 problems, and the authors conduct extensive experiments across over 20 language models, ensuring robust evaluation.

Clarity:
The paper is mainly well-organized, with clear explanations of the benchmark’s design, tasks, and experimental results.

Significance:
ERRORRADAR provides a critical tool for advancing MLLM performance in educational contexts, offering a realistic challenge that highlights current model limitations.

**Weaknesses:**

References: Key references are missing, undermining the paper’s scholarly foundation. For example, *MMMU: A Massive Multi-discipline Multimodal Understanding and Reasoning Benchmark for Expert AGI*, a relevant multimodal benchmark is omitted and should be cited in the discussion of related work. It includes collage level mathematics questions and has a similar error classification as your paper. Besides, the mention of unrelated fields, such as e-commerce and urban planning (P1 L38-39), distracts from the focus on educational applications and should be removed.

Error Analysis: While the error analysis identifies issues and possible explanations, it would be more valuable if accompanied by actionable suggestions. For instance, in Finding #4 within the main results, the authors could propose adding domain-specific knowledge to the dataset of foundation model as a possible solution. The improvement suggestions are only observed in 3.3.5 Visual Perception Case Study, which makes the analysis feel limited in scope. Without concrete recommendations throughout, the analysis lacks depth and does not contribute substantially to advancing practical solutions in the field.

Writing and Structure: The paper’s structure is suboptimal. Placing the related work after the introduction would provide a clearer flow, allowing readers to understand the motivations and contributions of ERRORRADAR in context with prior work. Specific points also lack consistency, such as the mixed use of "Obs.2" and "Finding #2" in P9 L455 and P7 L349, which should be standardized.

Figure Presentation: The figures are challenging to interpret and require considerable improvement. For instance, P6 Figure 5 lacks clarity, especially in differentiating the small section labeled "other" for miniCPM. The pie chart format does not effectively convey comparisons between models; a diverging bar chart could better illustrate the "CAL" percentage. Additionally, the color scheme is confusing, with the inner pie chart darker than the outer one with no meaning. It is also unnecessary, as the values are reflected in the size of the pie chart. Similarly, the images of questions in P10 Figure 10 suffers from low resolution, making the table at the central and other details difficult to read. Also, the organization of 5 cases can be improved, for example, they can be presented separately. The quality and readability of both figures should be substantially enhanced.

**Questions:**

To make this paper more solid and have more potential impact on the LLMs society, could you please address the issues in the weakness?

---

### Meta-Review · Area_Chair_4UMm · 2024-12-18

**Metareview:**

This was a borderline paper that I believe was submitted too soon.  The reviewers pointed out extensive issues with the paper and while the authors did significant work in addressing those issues during the rebuttal period, the main result was that one reviewer moved from marginally below to marginally above threshold.

There has been extensive reviewer analysis and author response, but to summarize, the paper simply needs more work, both isn terms of establishing the robustness of the benchmark and in the writing and presentation.  This paper could possibly be bumped up given all the authors work in the rebuttal period, but I really think it would be better if they re-wrote it and submitted again.  The requested corrections are extensive and have not been properly reviewed by all reviewers.

**Additional Comments On Reviewer Discussion:**

There were a lot of criticisms from the reviewers and the authors put in a tremendous amount of effort in the responses.  One reviewer replied that they had read the authors responses but were still not satisfied, prompting another round of response from the authors to which that reviewer did not respond.  The authors are complaining about the lack of response to their second round rebuttal from this reviewer, but I think that they are just not convinced.  Another reviewer went through two full rounds of discussion with the author and in the end in appreciation for all their efforts raised their score from a 5 to a 6.  Another reviewer who gave it a 6 despite raising a number of issues did not respond to the authors rebuttal.

---

### Decision · Program_Chairs · 2025-01-22

Reject